# Statistics of modal condensation in nonlinear multimode fibers

**Mario Zitelli** [1] ✉, **Fabio Mangini**[1] **& Stefan Wabnitz** [1]

Optical pulses traveling through multimode optical fibers encounter the influence of both linear disturbances and nonlinearity, resulting in a complex and chaotic redistribution of power among different modes. In our research, we explore the phenomenon where multimode fibers reach stable states marked by the concentration of energy into both single and multiple sub-systems. We introduce a weighted Bose-Einstein law, demonstrating its suitability in describing thermalized modal power distributions in the nonlinear regime, as well as steady-state distributions in the linear regime. We apply the law to experimental results and numerical simulations. Our findings reveal that, at power levels situated between the linear and soliton regimes, energy concentration occurs locally within higher-order modal groups before transitioning to global concentration in the fundamental mode within the soliton regime. This research broadens the application of thermodynamic principles to multimode fibers, uncovering previously unexplored optical states that exhibit characteristics akin to optical glass.

Multimode (MM) optical fibers[1–3] have experienced a resurgence in interest over the last decade. This renewed enthusiasm is driven by the potential to enhance transmission capacity through spatial-division multiplexing (SDM) techniques[4,5], as well as the opportunity to scale up the pulse energy delivered by fiber lasers[6].

In recent years, a thermodynamic interpretation of beam propagation in multimode optical fibers and systems has been developed[7,8]. To illustrate this concept using the example of a graded-index (GRIN) MM fiber, photons are considered as indistinguishable energy packets. The equilibrium distribution of these energy packets among degenerate mode groups, characterized by the same propagation constant $\beta_j$ ($m^{-1}$), can be estimated using statistical mechanics. This equilibrium distribution corresponds to the points of extrema of the entropy $S$ associated with the mode population. This thermodynamic perspective offers a significant simplification for the analysis of complex multimode systems and fibers, providing a powerful tool for designing multimode fiber transmission systems affected by disorder and nonlinear modal interactions.

Conversely, it is widely recognized that when optical pulses propagate within a MM fiber in the linear regime, they are subject to the influence of random-mode coupling (RMC)[9–11], which arises from imperfections in the fiber, such as micro and macro-bending. In consequence, pulses carried by modes belonging to degenerate groups, having either identical or very similar propagation constants, become temporally separated due to inter-modal dispersion. This separation leads to the broadening of spatiotemporal pulses carried by each group, primarily due to intra-modal or chromatic dispersion (see Fig. 1a).

In the linear regime, the effects of RMC can be effectively described by power-flow equations[9,12,13], which predict the diffusion of energy from intermediate mode groups into both lower-order and higher-order modes. These models, when applied to several hundred meters of fiber propagation, result in a steady-state mode power distribution characterized by a gradual decrease in power as the mode order increases.

At higher pulse energies, within MM fibers exhibiting anomalous chromatic dispersion, a distinctive optical phenomenon emerges: the formation of optical solitons with a specific pulse width denoted as $T_{FWHM}$. The width of these solitons, $T_{FWHM}$, depends on the pulse's wavelength, as described in ref. 14. The soliton energy, denoted as $E_s$,

[1]Department of Information Engineering, Electronics and Telecommunications, Universitá degli Studi di Roma Sapienza, Via Eudossiana 18, Rome 00184 RM, Italy. ✉e-mail: mario.zitelli@uniroma1.it

can be calculated using the equation:

$$E_s = \frac{1.76\lambda|\beta_2(\lambda)|w_e^2}{n_2 T_{\text{FWHM}}}. \qquad (1)$$

Here, $\lambda$ represents the pulse wavelength, $\beta_2(\lambda)$ (s²/m) the chromatic dispersion, $n_2$ (m²/W) the Kerr nonlinear coefficient, and $w_e$ the modal effective waist. At the telecom standard of $\lambda = 1550$ nm, $T_{\text{FWHM}}$ corresponds to a pulse width of 120 fs. As the pulse energy increases, pulses associated with different modal groups undergo a reduction in width while their peak power intensifies. This phenomenon occurs due to the influence of Kerr and Raman nonlinearities, which gradually transfer energy from each group to the fundamental mode[15]. This energy redistribution process involves a gradual condensation of energy into the fundamental mode. As pulses become temporally separated, the mode coupling process is sustained by the interplay of random-mode coupling (RMC) and inter-modal four-wave mixing (IM-FWM). After traveling over hundreds of meters within the fiber, what remains is a succession of fundamental soliton bullets. These solitons experience Raman soliton self-frequency shifting[16,17], as depicted in Fig. 1c. This intricate process can be viewed as a fission mechanism mediated by modal dispersion, and it will be further elucidated in this work.

In the intermediate energy range, typically spanning from 20% to 80% of the soliton energy $E_s$, a fascinating optical phenomenon occurs: the emergence of quasi-solitons. These quasi-solitons are characterized by individual pulses carried by distinct modal groups within the fiber. Initially, these pulses overlap, and their pulse widths approach the value observed in solitons. The primary mode of power exchange in this regime is governed by IM-FWM, and the Raman self-frequency shift does not significantly alter the pulse wavelength. Consequently, we can still consider the optical wave propagation as a conservative system. In this regime, the modes exhibit a characteristic distribution, characterized by the local condensation of energy among the lower-order mode groups (see Fig. 1b). This observed local condensation phenomenon within MM fibers bears similarities to phenomena observed in disordered lasers[18] and Bose-Einstein condensation[19]. It showcases the intricate and rich behavior that can arise in optical systems under specific conditions and energy regimes. The quasi-solitons addressed in this work should not be confused with the quasi-soliton pulses emitting Cherenkov radiation in single-mode fiber[20].

In the upcoming sections, we will delve into the demonstration of a weighted Bose-Einstein (BE) equation, which corresponds to the extrema of the entropy within an optical multimode system. Our exploration will encompass experiments conducted over extended lengths of graded-index (GRIN) multimode fibers, spanning from the linear regime to the soliton regime. We will scrutinize the modal content emerging from the fiber using the weighted BE equation to determine the attainment of either local or global condensation states.

We will conduct numerical power-flow simulations within the linear regime, followed by an analysis using the weighted BE equation. Furthermore, we will employ numerical simulations involving coupled-mode generalized nonlinear Schröedinger equations (GNLSE). These simulations will incorporate nonlinearity and an original model for RMC, enabling a comparison with experimental data.

The weighted BE equation will prove to be highly accurate in describing soliton condensation in the anomalous dispersion regime and self-cleaned thermalized states within the normal dispersion region. It will also exhibit good accuracy in depicting steady-state distributions within the linear regime. In Note A of the supplementary material, we will perform a comparison between the weighted BE equation and the well-known Rayleigh-Jeans law (RJ)[7,8], particularly when analyzing thermalized states. The former, although more complex to fit, will demonstrate superior accuracy. Furthermore, we will discover new steady states in the quasi-soliton regime. These states will be characterized by local condensates, akin to a glassy state[21,22]. In this context, a glassy state refers to a condition where the energy becomes localized within modal groups distinct from the ground state. This condition will manifest itself as an intermediate state between the low-energy disordered state and the high-energy, highly condensed soliton state.

## Results
### Weighted Bose-Einstein Law
We commence our analysis by considering an optical multimode system comprising $Q$ groups of degenerate modes, each distributed across $g_j/2$ modes and encompassing two polarizations ($g_j$ represents the group degeneracy, with $j = 1, 2, \ldots, Q$). In the specific case of a GRIN multimode fiber, the degeneracy values are $g_j = 2, 4, 6, \ldots, 2Q$. Within this context, we denote by $n_j$ the population of energy packets associated with the $j$-th group, accounting for both polarizations. In practical applications, the number of indistinguishable energy packets involve values of $n_j$ ranging from approximately $10^5$ to $10^9$, corresponding to the number of photons in a fiber modal group.

To derive the optimal modal power fractions within the system, we employ an entropy extremization approach, incorporating the appropriate Lagrange multipliers. This leads us to the weighted Bose-Einstein distribution, which characterizes the distribution of power among different modes (for detailed methodology, see Section Theory).

$$|f_i|^2 = \frac{2(g_i - 1)}{g_i \gamma} \frac{1}{\exp\left(-\frac{\mu' + \epsilon_i}{T}\right) - 1}. \qquad (2)$$

In Equation (2), the quantity $|f_i|^2$ represents the mean modal power fraction across two polarizations. It is calculated as $2n_j/(\gamma n_0 g_i)$, where $\gamma$ corresponds to the total number of energy packets $N$ as $\gamma = N/n_0$, and $n_0$ serves as a reference number of packets, often taken as the value at the lowest tested power. The index $i$ refers to the mode within the $j$-th group. For a GRIN fiber, there are $2M = Q(Q+1)$ modes and polarizations, and $i$ takes on values such as $i = 1$ for $j = 1$, $i = 2, 3$ for $j = 2$, $i = 4, 5, 6$ for $j = 3$, and so forth. The values of $g_i$ are given by the sequence $[2, 4, 4, 6, 6, 6, \ldots, 2Q]$, and $\epsilon_i = \beta_i - \beta_{j=Q}$ represents the differential modal eigenvalues, with $\beta_i$ (m⁻¹) being the propagation constants calculated according to Equation 16 in ref. 13.

Equation (2) is derived under the reasonable condition that $n_0$ is significantly larger than $\exp(-\frac{\mu' + \epsilon_i}{T}) - 1$. This equation can effectively

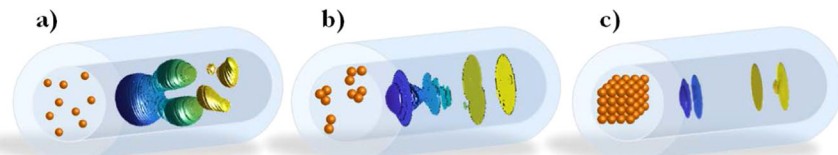

**Fig. 1 | Optical modal bullets that emerge following extended propagation through graded-index (GRIN) fiber. a** Linear, **b** nonlinear, and **c** soliton regimes. Blue (yellow) bullets correspond to the propagated lower-order (higher-order) modes. The three thermodynamic states (gas, glassy, solid) are creatively illustrated in orange color.

replicate the modal power distribution both in the linear regime, where RMC primarily governs power exchange among modes, and in the nonlinear regime, where inter-modal four-wave mixing (IM-FWM) takes precedence.

The chemical potential $\mu'$ (m$^{-1}$), temperature $T$ (m$^{-1}$), and the normalized power $\gamma$ constitute the three degrees of freedom for fitting Equation (2) to experimental data. The only constraint imposed on $\gamma$ is that it scales with the input power while also adhering to the conservation law $\sum_{i=1}^{M} |f_i|^2 = 1$.

The accuracy of the thermodynamic approach represented by Equation (2) can be checked by considering the state equation presented in Section Theory, specifically Equation (19). This equation can be reformulated as follows:

$$SE = \sum_{j=1}^{Q} \beta_j j |f_j|^2 + \mu' - \beta_{j=Q} + (2M - Q)\frac{T}{\gamma} = 0; \qquad (3)$$

The calculation of an experimental error related to the state equation can be expressed as:

$$\epsilon_{SE} = \frac{SE}{\sum_{j=1}^{Q} \beta_j j |f_j|^2 - \left(\mu' - \beta_{j=Q} + (2M - Q)\frac{T}{\gamma}\right)}. \qquad (4)$$

In Equation (3), it's essential to note that the first term in the sum, which represents the normalized energy, must remain constant. The parameters $\mu'$ and $T/\gamma$ must be scaled to maintain the remaining part of the equation constant. It's not mandatory for the parameters $T/\gamma$ and $\mu'$ to remain individually constant.

In cases involving short fiber spans and low power levels, where both RMC and IM-FWM are negligible, we can assume that the modal distributions at the input and output of the fiber are equal, i.e., $|f_j^{(in)}|^2 = |f_j^{(out)}|^2$. Given that the input modal distribution remains unchanged at high power, we can use Equation (3), along with $|f_j^{(out)}|^2$ measured at low power, to predict the thermodynamic parameters of the system at high power when thermalization or condensation is achieved[23].

However, in the case of long spans of fiber at low power, RMC introduces a discrepancy such that $|f_j^{(in)}|^2 \neq |f_j^{(out)}|^2$. In such scenarios, thermodynamic parameters are determined by fitting Equation (2) to the experimental distribution at a specific power level. While Equation (3) can be used to validate the thermodynamic consistency of the fit, it is not employed to predict parameters from low-power experiments.

The validity of Equation (2) is also contingent on the constraints of negligible variations in power $P$ and internal energy $U$, as outlined in the Method Section Theory. Linear losses are generally negligible at the telecom wavelength for fiber lengths up to a few kilometers. Raman nonlinearity causes a frequency shift in the pulse spectrum, leading to variations in packet energy. In soliton propagation, Raman self-frequency shift (RSFS) induces a red-shift in the spectrum without distortion; the propagation constant of the soliton changes accordingly, and so does the Hamiltonian. Assuming a 5% tolerance on the internal energy change, an 80 nm red-shift at a 1550 nm wavelength can be tolerated in the experiments.

Chromatic dispersion significantly broadens pulses in the linear regime, reducing the pulse peak power $P$. However, in the quasi-soliton and soliton regimes, the propagating pulse train retains its pulse width, and negligible power changes can be assumed over substantial fiber lengths.

RMC also introduces negligible energy variations. For instance, consider propagation at 1550 nm in a GRIN fiber, where RMC leads to a complete power transfer from group 10 to the fundamental mode. This represents a worst-case scenario, and even in such instances, the fractional energy change is approximately $(\beta_1 - \beta_{10})/\beta_1 = 0.0078$. In practical scenarios, strong RMC results in internal energy variations

that are negligible and comparable to the linear losses over a few meters of fiber. Consequently, we can assume that the presence of strong RMC does not invalidate the thermodynamic approach.

Experimental modal distributions that are well-fitted by Equation (2) correspond to the extrema of the entropy $S$, indicating the achievement of a steady state, which may be characterized by either local or global condensation. Power fluctuations related to local condensation do not invalidate the equation, as it is used to fit the power distribution of the modal groups as a whole. In the following section, Equation (2) will be applied for this purpose.

## Experiments

To determine the mode power distribution at the output of GRIN MM fibers, we employed the mode-decomposition method introduced in ref. 24. Our experimental setup is elaborated upon in Section Experimental setup. We systematically varied the input pulse energy $E_{in}$, allowing us to investigate the full spatio-temporal propagation regime, ranging from linear to nonlinear cases. We utilized 250 fs full-width-at-half-maximum (FWHM) pulses at 1400 nm wavelength, generated with a repetition rate of 100 kHz. The input beam was circularly polarized and coupled with a waist of 13 μm, introducing a 10 μm lateral shift relative to the fiber axis. This configuration was designed to enhance the population of higher-order modes (HOMs) while minimizing power exchange between polarizations. Our experiments employed commercial OM4 GRIN fibers (Thorlabs GIF50E) with lengths of 1 m, 830 m, and 5 km, respectively, spooled on a support structure with a radius of curvature greater than 8 cm.

Figure 2 presents the normalized instantaneous output power (left) and the near-field (right) after 830 m of GRIN fiber. The observed time delay among pulses carried by different groups of degenerate modes is a consequence of modal dispersion. It's noteworthy that the sub-pulses exhibit equal temporal spacing, owing to the uniform spacing of mode propagation constants in GRIN fibers. The pulse carried by the HOMs experiences the greatest delay, causing it to appear in the trailing portion of the output waveform.

Employing a sufficiently long fiber to temporally separate different mode groups allows us to directly measure the output mode power distribution. This distribution arises from the combined effects of linear and nonlinear mode coupling. The nonlinear interactions among different modal groups are primarily concentrated in the initial segment of the fiber, where pulses temporally overlap. Consequently, we can assume negligible linear losses over the interaction length, enabling a valid thermodynamic interpretation of the results. While the linear interaction is indeed significant over the entire length of fiber propagation, as mentioned in Section Weighted Bose-Einstein Law, it's important to emphasize that it does not undermine the validity of the thermodynamic approach, in the cases where the peak power of pulses is not affected. The thermodynamic model remains applicable and provides valuable insights even in the presence of substantial linear interactions.

In the linear regime, we estimate that chromatic dispersion (CD) causes individual pulses within each group to broaden up to a duration of 110 ps after traveling 830 m, considering the nominal chromatic dispersion of $-12$ ps$^2$/km. Modal dispersion introduces a time delay of 206 ps between adjacent group pulses.

Our experiments involve varying the optical pulse energy from the linear regime (0.1 nJ) to the soliton regime (4.5 nJ). The quasi-soliton regime is achieved within the energy range of 1.0 to 4.1 nJ, characterized by the presence of trains of short pulses. The Raman soliton regime is reached at 4.5 nJ, with no observed Raman delay or spectral shift up to 4.1 nJ. Therefore, we assume a conservative system behavior up to 4.1 nJ.

In the linear regime (as shown in Fig. 2a), the distribution of energy among the mode groups at the output is primarily determined

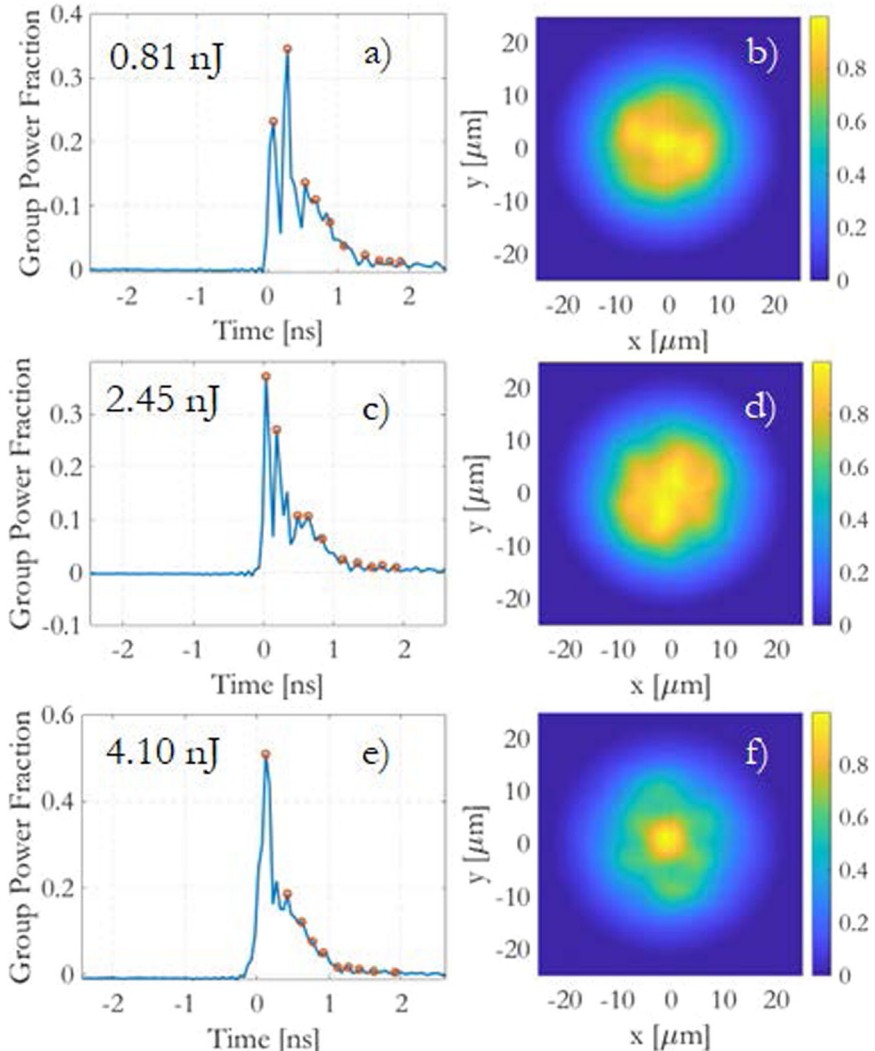

**Fig. 2 | Measured photodiode traces and near-fields.** (Left) normalized traces sampled at a 206 ps interval to yield group power fractions. The corresponding near-field intensities (right) are also shown after 830 m of GRIN fiber, considering input pulse energies of: **a**, **b**) 0.81 nJ, **c**, **d**) 2.45 nJ, **e**, **f**) 4.10 nJ.

by the linear disorder. For an input energy of 0.81 nJ, the effective length of the fiber, considering both weak linear losses and the rapid decrease in pulse peak power due to chromatic dispersion-induced temporal broadening, is as short as $L_{eff} = 12$ m. This implies that, for most of the distance, power exchange among pulses carried in different modes is predominantly influenced by RMC.

In Figure 2a, c, e, we sample the photodiode traces at points corresponding to the relative group delay of each modal group (indicated by the orange circles). This allows us to perform a mode power decomposition, as described in ref. 24. In this approach, considering the first $Q = 10$ modal groups, which corresponds to $M = Q(Q+1)/2 = 55$ modes per polarization, we directly measure the mode group powers $P_j$. Subsequently, we calculate the mean modal power fraction carried by each mode within the group as $|f_i|^2 = 2P_j/(g_i P_{tot})$, assuming power equipartition within each group (for detailed methodology, see Methods Section Power-flow numerical model).

In the linear propagation regime (i.e., for $E_{in} \leq 0.81$ nJ), Fig. 2a illustrates that most of the pulse energy is primarily carried by the first 6 mode groups, particularly concentrating in the second group, under the experimental coupling conditions.

Conversely, as depicted in Fig. 2c, for $E_{in} = 2.45$ nJ, lower-order modes start attracting power from the HOMs due to the increasing influence of IM-FWM. This energy level corresponds to a strongly nonlinear quasi-soliton regime.

At 4.10 nJ pulse energy (Fig. 2e), the fundamental mode has captured half of the total power, and the propagation is approaching a multimode soliton. However, the soliton has not fully formed and lacks the characteristic Raman delay of a complete soliton regime[14]. For this reason, we can disregard the presence of Raman scattering and other dissipative effects up to approximately this level of input pulse energy.

In our experiments, we also observed that for $E_{in} \geq 4.6$ nJ (not shown), a soliton forms in the fundamental mode. With higher values of $E_{in}$, the soliton becomes further delayed in time and separates from the remaining pulse carried by the HOMs, owing to the Raman soliton self-frequency shift[16,17,25,26].

As depicted in Fig. 2b, d, the output mode power distribution yields a relatively broad and speckled output beam intensity profile at both low and intermediate values of $E_{in}$. However, as we approach the soliton regime, where most of the energy is attracted by the fundamental mode, Fig. 2f shows that the beam brightness significantly increases at its center. This results in the formation of a bell-shaped beam with a waist close to that of the fundamental mode, surrounded by a background of HOMs. Such a beam profile is a characteristic signature of beam self-cleaning, where the fundamental mode dominates and effectively eliminates the influence of other modes, leading to a cleaner and more focused output beam[27].

Figure 3 provides a log-scale representation of the average output power fractions $|f_i|^2$ as a function of their corresponding mode

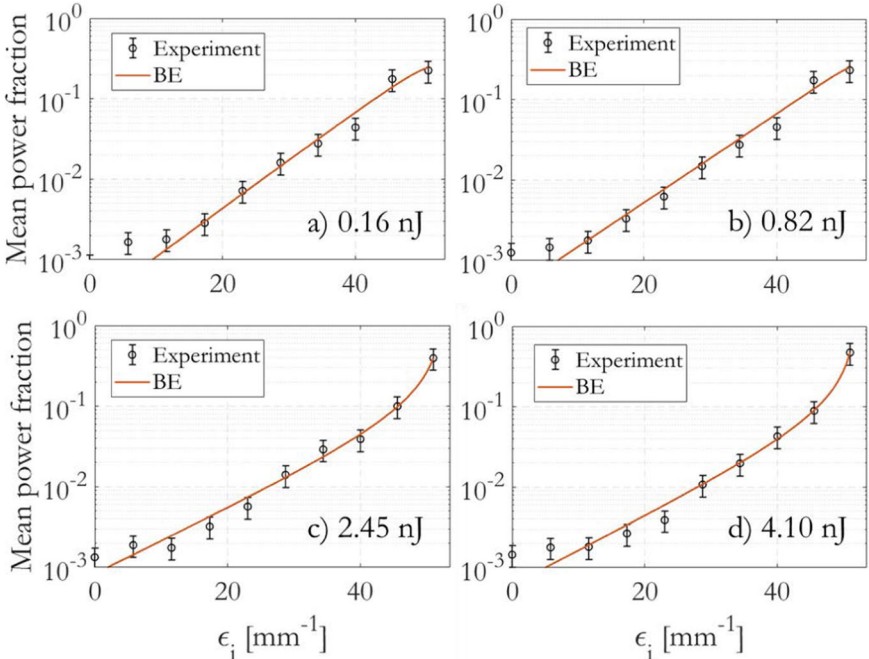

**Fig. 3 | Average mode power fraction $|f_i|^2$ (black circles) as a function of modal eigenvalue $\epsilon_i$ for various input pulse energies $E_{in}$. a** 0.16 nJ; **b** 0.82 nJ; **c** 2.45 nJ; **d** 4.10 nJ. Error bars corresponding to an estimated 15% error are added. In each plot, you can observe the corresponding fits obtained using the weighted BE distribution, represented by the solid orange curve. These fits provide a good match to the experimental data and demonstrate the applicability of the weighted BE distribution in describing the mode power distribution for different input pulse energies.

eigenvalues $\epsilon_i$, using the accurate decomposition method. The input pulse energy in Fig. 3 spans from $E_{in} = 0.2$ nJ, where mode mixing is primarily influenced by RMC, to $E_{in} = 4.1$ nJ, where IM-FWM becomes the dominant mechanism for transferring energy among non-degenerate modes. This comprehensive range of input energies allows us to observe the evolution of the mode power distribution as different nonlinear effects come into play[27].

The output modal distributions are fitted using Eq. (2), ensuring that the constraint for the normalized power $\gamma$ is respected. In particular, $\gamma$, $T$, and $\mu'$ are initially determined from the fit at an intermediate pulse energy. For other energy values, $\gamma$ scales with the pulse energy $E_{in}$, while $T$ and $\mu'$ are determined from the fits.

At low pulse energy (Fig. 3a), the weighted BE distribution approximates a straight line, which is in good agreement with predictions obtained by numerically solving the power-flow equations[9,12]. This agreement highlights the validity of the Bose-Einstein law in describing the mode of power distribution in the linear regime (see Sec. Linear disorder).

For a pulse energy of 0.81 nJ (Fig. 3b), the propagation is initially nonlinear and later becomes dominated by RMC. The weighted Bose-Einstein distribution properly fits the experimental data up to the 9th group order, with $\mu' = -58.2$ mm$^{-1}$, $T = 8.71$ mm$^{-1}$, and $\gamma = 3.60$. In the figure, it can be observed that group 2 has a larger energy fraction compared to the fitting equation. It will be shown later that at this power level, there is a local condensation of energy into lower groups.

On the other hand, Fig. 3c, d show that as soon as $E_{in} \geq 2.45$ nJ, the population of the fundamental mode increases, and it preferentially acquires power from HOMs as a consequence of IM-FWM. In both cases, the weighted BE distribution fits the experimental data up to the soliton regime, with $\mu' = -53.1(-52.2)$ mm$^{-1}$, $T = 8.71(10.14)$ mm$^{-1}$, and $\gamma = 10.8(18.0)$ for $E_{in} = 2.45(4.10)$ nJ, respectively. The BE distribution represents a new equilibrium state induced by strong nonlinearity, which accumulates over the entire fiber length.

It's worth noting that for $E_{in} = 4.1$ nJ, $(\mu' + \epsilon_1)/T \simeq -0.13$, indicating that the RJ approximation to the BE law (i.e., $|f_i|^2 \propto -T/(\mu' + \epsilon_i)$) is appropriate around the fundamental mode with $i = 1$.

Figure 3 in Note B of the supplementary material overlaps the measured power fractions in linear scale, for a direct comparison at different energy levels.

Table 1 presents the values of the optical temperature $T$, differential potential $\mu'$, factor $\gamma$, and state equation error $\epsilon_{SE}$ for the experiment depicted in Fig. 3. As the input energy $E_{in}$ increases, the temperature $T$ rises from 7.11 mm$^{-1}$ to 10.36 mm$^{-1}$. Correspondingly, $\mu'$ decreases from $-65.42$ mm$^{-1}$ to $-52.08$ mm$^{-1}$. The factor $\gamma$ scales proportionally to the input energy, and the error in the state equation decreases from 7.9% and 3.2% in the linear regime to 0.3% in the nonlinear regime. These variations in the thermodynamic parameters highlight the changing behavior of the multimode system as it transitions from linear to nonlinear regimes. The r-squared figure of merit for the fits increases with energy from 0.954 to 0.999, demonstrating excellent accuracy for the weighted BE distribution when fitting thermalized states, and acceptable accuracy for the linear steady-states. This indicates that the weighted BE provides a reliable description of the mode power distribution over a wide range of input energy levels, capturing both the thermalization process and the steady-state behavior in multimode optical fibers.

Let us now provide additional information on how the relative power fractions for groups of nondegenerate modes, or $j \cdot |f_j|^2$, evolve with input energy. Figure 4a, b illustrate the behavior of these power fractions as a function of group index $j$ for different input energies. This provides a clear visual representation of how energy is distributed among different mode groups as the input energy increases and the system transitions from linear to nonlinear regimes.

In Fig. 4a (experimental data), it's interesting to observe local and global condensation phenomena as the input energy increases. When passing from 0.16 nJ to 0.82 nJ input energy, the power of group 2 increases, followed by an important decrease for 1.65 nJ of input energy or higher values. At the intermediate energies of 0.82 nJ there is local condensation into group 2 at the expense of HOMs. At higher energies (1.65 nJ and 3.29 nJ), local condensation into group 4 is observed. Finally, as the system approaches the soliton regime (4.61 nJ), the fundamental mode gains a larger share of

the total output pulse energy, at the expenses of groups 2-4 and HOMs.

In Fig. 4b (numerical simulations), a similar trend is observed. At 1.0 nJ, there is local condensation in group 2. At 3.0 nJ, there is local condensation in group 3. Finally, at 5.0 nJ, there is global condensation in group 1.

These results provide valuable insights into how energy distribution among different mode groups evolves in multimode optical fibers as the input energy and nonlinearity increase. They also demonstrate the utility of the weighted BE distribution in describing these phenomena. Additionally, the validation against independent experimental data[23] in the supplementary section further supports the accuracy of the approach.

## Simulations

**Linear disorder.** In the linear propagation regime of multimode optical fibers, modal power exchange primarily originates from RMC. In multimode optical fibers, after a certain propagation distance represented by $z_{SSD}$, RMC alone can produce a steady-state output mode power distribution (SSD). This SSD arises as a result of the power exchange among different modes within the fiber due to RMC. Importantly, once this steady-state output mode power distribution is achieved, further propagation beyond $z_{SSD}$ does not significantly affect the modal distribution.

**Table 1 | Thermodynamic parameters obtained by fitting the data from Fig. 3 using the weighted Bose-Einstein distribution**

| $E_{in}$ [nJ] | $T$ [mm⁻¹] | $\mu'$ [mm⁻¹] | $\gamma$ | $\epsilon_{SE}$ | $R^2$ |
|---|---|---|---|---|---|
| 0.16 | 7.11 | − 65.42 | 0.72 | 0.0793 | 0.954 |
| 0.41 | 7.76 | − 60.92 | 1.80 | 0.0318 | 0.951 |
| 0.82 | 8.71 | − 58.19 | 3.60 | 0.0177 | 0.949 |
| 1.23 | 9.32 | − 56.52 | 5.40 | 0.0122 | 0.962 |
| 1.64 | 9.05 | − 54.74 | 7.20 | 0.0085 | 0.984 |
| 2.06 | 9.53 | − 54.17 | 9.00 | 0.0069 | 0.987 |
| 2.45 | 8.71 | − 53.10 | 10.80 | 0.0058 | 0.990 |
| 3.29 | 9.75 | − 52.81 | 14.40 | 0.0041 | 0.991 |
| 4.11 | 10.14 | − 52.29 | 18.00 | 0.0035 | 0.998 |
| 4.61 | 10.36 | − 52.08 | 20.16 | 0.0035 | 0.999 |

The errors on the state equation ($\epsilon_{SE}$) and the figure of merit ($R^2$) of the fits for the data are also reported. These values indicate the accuracy of the fits using the weighted BE distribution in describing the experimental data. As the input energy increases and the system transitions to a nonlinear regime, both the error on the state equation and the figure of merit of the fits improve, indicating a closer agreement between the model and the experimental results.

Power-flow equations are a well-known model used to simulate the linear coupling among modes in a GRIN optical fiber. These equations describe the power exchange and propagation of optical modes within the fiber, taking into account factors like modal dispersion, coupling between modes, and other linear effects. They are a set of differential equations that help predict how the power carried by different modes changes as the optical signal travels through the fiber.[12],[9] (see Sec. Power-flow numerical model). The power-flow equations involve a bi-directional flow of power among different mode groups within the optical fiber. This flow of power can occur in both directions, from group $j$ to group $j − 1$ and from group $j$ to group $j + 1$, allowing for energy transfer between adjacent mode groups. Over multiple integration steps, this process can lead to a cascading effect, where power flows into non-adjacent mode groups as well.

One key aspect of these equations is that the mode coupling coefficients, which govern the power transfer between different mode groups, are not necessarily symmetrical in both directions. This property applies among groups, and does not apply to the individual modes into groups. As discussed in Methods Sec. Power-flow numerical model, this lack of symmetry can result in a preferential transfer of power from HOMs into lower-order modes within the optical fiber.

The numerical solution of the power-flow equations in Fig. 5 provides a visual representation of how power propagates among the first 10 mode groups in a multimode optical fiber. This simulation considers a specific scenario with the same coupling conditions of the experiment in Fig. 2, which involves the following parameters: wavelength: 1400 nm; fiber length: 830 m of GRIN fiber; linear loss: 2.6 dB/km; coupling coefficient: $D = 0.01$ m⁻¹ for a glass GRIN fiber with bending[28,29]; we also included weak modal losses: $A = 1 \times 10^{-4}$ m⁻¹. The simulation shows how power is transferred between these mode groups as they propagate through the fiber.

The simulation and the experiment of Fig. 5 provide valuable insights into the power distribution and mode coupling dynamics in the multimode optical fiber. Figure 5a is a simulation showing how the power of different mode groups evolves as they propagate through the optical fiber. The net flow of power is in the direction of lower-order mode groups, indicating that power is transferred from higher-order modes to lower-order modes as they travel through the fiber. This is a characteristic feature of RMC in multimode fibers, when HOMs are stimulated at the input end. Figure 5b compares the simulation with the experiment of Fig. 3a at 0.16 nJ input energy; it illustrates how the mean modal power fraction $|f_i|^2$ changes as a function of the mode eigenvalue. As power flows from higher-order modes to lower-order modes, the mean modal power fractions increase for the lower-order modes, indicating that they accumulate more power during propagation. Figure 5b also reports the simulation with no modal losses ($A = 0$),

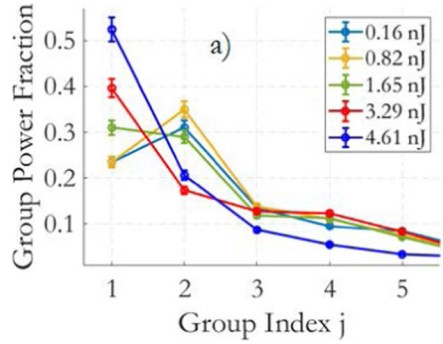

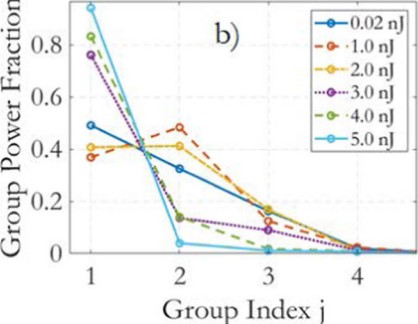

**Fig. 4 | Normalized power of the groups of modes. a** Experimental group power fraction $j \cdot |f_j|^2$ plotted against the group index $j$ for input pulse energies ranging between 0.16 nJ and 4.61 nJ. **b** Simulated simulated group power fraction under similar conditions to the experiments. One notable observation is that local power

condensation is observed in groups 2–4 at intermediate energy levels. This phenomenon suggests that energy condensation occurs in these lower-order groups as the pulse energy increases, which is a characteristic behavior of the multimode system under these conditions.

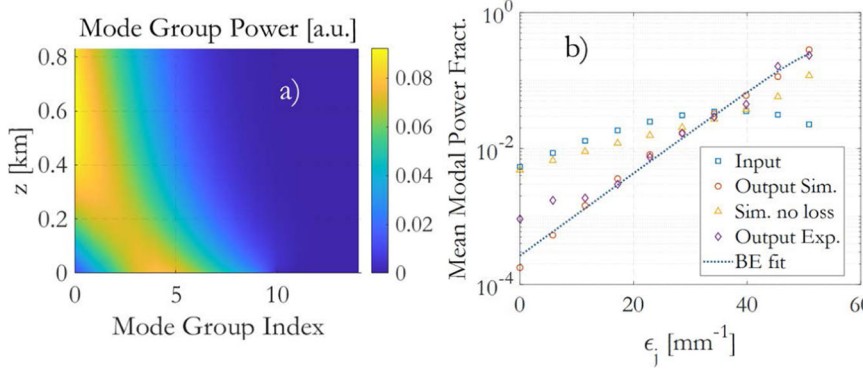

**Fig. 5 | Power-flow simulation in the linear regime. a** Power of the mode groups vs. distance, and **b** corresponding input and output mean modal power fractions vs. the modal eigenvalues, compared to the experiment; simulation with no modal losses is also included (Sim. no loss). The good agreement between the simulated and experimental data in terms of the mean modal power fractions further validates the use of the weighted BE law in describing the steady-state mode power distribution in the linear regime.

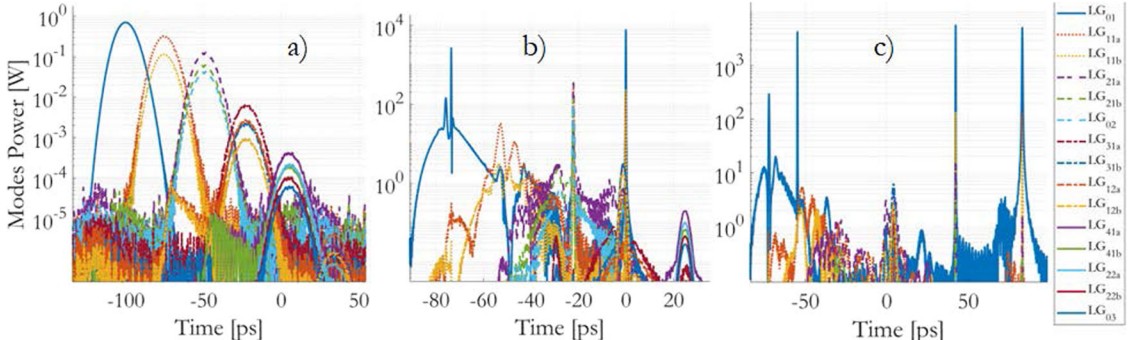

**Fig. 6 | Coupled-mode GNLSE simulations.** Output modal power for $E_{in}$ equal to **a** 0.02 nJ, **b** 3.0 nJ and **c** 5.0 nJ, respectively, with the same input coupling conditions as in the experiment.

demonstrating that RMC alone is able to promote the lower-order modes. Modal losses eventually enhance the effect, further depleting the higher-order modes. The agreement between the power-flow simulation and the experimental data, as well as the successful fit using the weighted BE law, indicates that this approach is adaptable and suitable for describing the steady-state mode power distribution in the linear regime, especially up to the 8th modal group, provided the error on the state equation $\epsilon_{SE}$ is below a few percent. This demonstrates the effectiveness of the weighted BE law in capturing the dynamics of mode coupling and power flow in multimode optical fibers, even in complex and nonlinear regimes.

**Linear disorder and nonlinearity**. The use of coupled-mode GNLSEs, which include wavelength-dependent linear losses and an original model for RMC derived from power-flow equations[12], as explained in Method Sec. GNLSE numerical model, is a comprehensive approach to capture the experimentally observed intricate dynamics of mode coupling in multimode optical fibers[30]. This modeling strategy allows us to consider both linear and nonlinear effects in a physically accurate manner. These simulations can provide valuable insights into the mode coupling dynamics, power distribution, and energy transfer processes, helping to validate and extend the understanding of our experimental results. Modal losses were neglected in the simulations.

Our approach to save computation time by propagating a reduced number of modes (28 modes) over a shorter distance (100 m) of the GRIN fiber is a reasonable strategy for numerical simulations. This approach allows us to focus on a subset of relevant modes while still capturing key aspects of the mode coupling dynamics and

nonlinear effects. We considered an input 250 fs pulse at 1400 nm, with the same coupling conditions as in the experiments. Modes 1 to 28 correspond to the Laguerre-Gauss modes $LG_{01}$, $LG_{11e}$, $LG_{11o}$, $LG_{21e}$, $LG_{21o}$, ..., and $LG_{04}$, respectively.

Setting the RMC coupling coefficient and the RMC step appropriately is crucial for achieving accurate simulation results. Adjusting these parameters to match the physical characteristics of our specific fiber, permits us to obtain simulations that closely resemble real-world behavior. Specifically, we set the RMC coupling coefficient $D = 0.003$ m$^{-1}$, which introduces a significant amount of linear disorder over the considered distance. The RMC step was equal to $L_c = 6$ mm, to ensure appropriate simulation accuracy.

It's also noteworthy that we have considered typical parameters for an OM4 GRIN fiber at 1400 nm, such as chromatic dispersion, modal dispersion, Kerr and Raman nonlinearities, and linear losses. These parameters are essential for accurately modeling the behavior of optical pulses in MM fibers and ensuring that simulations provide meaningful insights into the observed phenomena.

The simulated results in Fig. 6 provide a clear visual representation of how the temporal dynamics of the output modal power change with varying input pulse energy in the linear, nonlinear, and soliton regimes. Here are some key observations: Linear Regime ($E_{in} = 0.02$ nJ, Fig. 6a): Pulses in each mode group experience broadening (up to 13 ps) due to chromatic dispersion. Pulses in different mode groups are separated in time by inter-modal dispersion. This regime shows typical linear behavior with minimal nonlinear effects. Nonlinear Regime ($E_{in} = 3.0$ nJ, Fig. 6b): Pulses within each mode group are significantly compressed in time. This compression is primarily due to self-phase modulation (SPM) and anomalous dispersion. Pulses do not exhibit

Raman delay, indicating that they haven't reached a full soliton regime yet. Power transfer occurs, with lower-order modes gaining energy through IM-FWM. Soliton Regime ($E_{in}$ = 5.0 nJ, Fig. 6c): Ultrashort Raman delayed soliton pulses (170 fs duration) are formed. A significant portion (nearly 90%) of the power is concentrated in the fundamental mode. This regime represents a state of highly condensed, stable soliton-like pulses.

The observation of a new fission mechanism in Fig. 6c is quite interesting. Here's a breakdown of this mechanism: Modal Dispersion Separation: As a consequence of modal dispersion, pulses within different mode groups are separated in time. This separation is due to the differences in group velocities of these modes. RMC-Induced Repopulation: The action of RMC has the effect of redistributing power among modes. It appears that RMC is causing some energy to be transferred to lower-order modes even in pulses that originally corresponded to high-order mode groups. This redistribution is indicated by the increased power in the lower-order modes. IM-FWM: after RMC has repopulated the fundamental and lower-order modes in all propagating pulses, the IM-FWM, a nonlinear process, boosts transfer of power from the HOMs to the fundamental mode into the individual pulses. Formation of Soliton-Like Pulses: After 100 m of propagation in the simulation, the combined effects of RMC, IM-FWM, and modal dispersion result in the formation of a train of fundamental solitons. These solitons are characterized by their stability and well-defined temporal characteristics. This mechanism highlights the complex interplay between linear and nonlinear effects in multimode fibers. The interaction of modal dispersion, RMC, and IM-FWM can lead to unexpected behavior, such as the formation of soliton-like pulses and the repopulation of lower-order modes within high-order mode-group pulses. Understanding these mechanisms is crucial for designing and optimizing optical fiber systems for various applications.

Results from the numerical simulations of Fig. 6 provide the mode group power fraction distribution $j \cdot |f_j|^2$ of Fig. 4b, which gives valuable insights into different regimes of multimode fiber propagation:

Linear Regime ($E_{in}$ = 0.02 nJ or 1.0 nJ): In the linear propagation regime, the group power decreases following a convex curve (in log scale). This behavior is indicative of the mode mixing effects primarily driven by RMC. The weighted BE law can accurately fit the output modal distributions in this regime, suggesting that a steady state is reached.

Intermediate Energy ($E_{in}$ = 2.0 nJ to 3.0 nJ): At intermediate energy levels, there is a departure from the expected linear decreasing shape of the group power curve. Instead, local energy condensations are observed into groups 2 and 3. This phenomenon could be a result of a complex interplay between linear and nonlinear effects, including RMC and IM-FWM.

Soliton Regime ($E_{in}$ = 5.0 nJ): When Raman solitons are formed, the group power decreases following a concave curve (in log scale). This curve shape is associated with nonlinear irreversible energy condensation into the fundamental mode, indicating thermalization. The weighted BE law can fit the output modal distributions in this regime, suggesting the achievement of a thermalized state.

These observations highlight the dynamic nature of mode power distribution in multimode fibers and how it evolves with input pulse energy. The ability of the weighted BE distribution to accurately fit the simulated output modal distributions in both linear and soliton regimes further underscores its applicability in describing steady states and thermalization in multimode fiber systems. For further details, we refer to Note C of the supplementary material (Fig. 4), which also reports the input modal distribution used for the simulations; by comparing the input and output distribution at low pulse energy, simulations confirm that lower-order modes are promoted by the RMC even in the absence of modal losses.

## Discussion

Our study provides a comprehensive understanding of the observed mode power distribution and condensation at the output of long lenghts of GRIN MM fibers, and their correspondence with the weighted BE distribution. Here's a summary of the key points:

### Steady states in linear, quasi-soliton, and soliton regimes

The mean modal power fractions $|f_i|^2$ show the achievement of steady states in the linear, quasi-soliton, and soliton regimes. The weighted BE law accurately fits the distributions reached after long distances, confirming the existence of steady states even at low power. This highlights the reliability of the weighted BE law in describing steady states and thermalization in multimode fiber systems.

### Power redistribution mechanisms

RMC scattering plays a significant role in redistributing power among modes. In the linear regime, RMC contributes to the formation of a BE distribution. However, in the quasi-soliton and soliton regimes, the Kerr nonlinearity and anomalous chromatic dispersion considerably shorten pulses; RMC diffuses power to the lower-order modes, a process which is boosted by the IM-FWM.

### Local energy attraction

Between 0.4 and 2.0 nJ input pulse energy, there is a local attraction of energy to lower-order groups (groups 2 and 3) at the expense of HOMs. Globally, thanks to the interplay of RMC and IM-FWM, energy clusters are formed in the lower-order groups in the quasi-soliton regime (Figs. 1b, 4 and 6b); a train of quasi-soliton pulses is produced, each composed by the modes of one group $j$, plus a fraction of modes belonging to the lower adjacent groups and of the fundamental mode; this results in the promotion of the first 3-4 modal groups at the output, and power clustering. In the framework of such interplay, RMC invalidates the achievement of a global condensation state as described by a RJ law[31]; however, the formation of steady states associated to a local condensation is still possible, a process which is characterized by a weighted BE modal distribution.

### Global modal condensation

In the soliton regime (Figs. 1c and 6c), global modal condensation to the ground state is observed in all splitted pulses; nearly 90% of the power is attracted to the fundamental mode, as a consequence of the interplay of RMC and IM-FWM. Pulse width reduces to 170 fs, which is typical of propagated walk-off solitons[14]. A train of fundamental solitons is eventually produced by a peculiar fission mechanism mediated by modal dispersion; then solitons are affected by Raman soliton self-frequency shift[17].

### Thermodynamic interpretation

The observed distributions can be interpreted in thermodynamic terms, where the linear regime corresponds to a gas-like state of energy packets. In the intermediate nonlinear regime, the system evolves into a locally condensed or "glassy" state. Finally, in the soliton regime, a globally condensed "solid" state is achieved[32].

Our detailed explanations provide a clear perspective on the complex interplay of physical mechanisms, and their impact on mode power distribution in multimode fibers at different power levels.

## Methods
### Theory

In the context of optical multimode systems, especially when dealing with a GRIN fiber, we often encounter a scenario where the modes are organized into $Q$ different groups. These groups are characterized by

their degeneracy, denoted as $g_j/2$, with $j = 1, 2, .., Q$, and typically involve two polarizations. In the special case of a GRIN fiber, it is $g_j = 2, 4, 6, . . . , 2Q$; the number of modes and polarizations is $2M = Q(Q + 1)$.

To better understand the distribution of energy packets within this system, a procedure outlined in ref. 7, can be followed. The population $n_j$ of energy packets into the j-th group over 2 polarizations, leads to a total number of combinations across the different groups and polarizations

$$W = \prod_{j=1}^{Q} \frac{(n_j + g_j - 1)!}{(g_j - 1)! n_j!}. \tag{5}$$

The multimode system entropy is defined as $S = ln(W)$; by applying the Stirling approximation, valid for $n_j + g_j - 1 >> 1$ we obtain

$$S = \sum_{j=1}^{Q} (n_j + g_j - 1)\Big[ln(n_j + g_j - 1) - 1\Big] - (g_j - 1)\Big[ln(g_j - 1) - 1\Big] \\ - n_j\Big[ln(n_j) - 1\Big]; \tag{6}$$

Global thermalization of the multimode optical system can be achieved by optimizing Eq. (6). This optimization process involves the use of Lagrange multipliers to account for the preservation of certain system properties. In this case, we are concerned with two conserved quantities: the total number of particles $N = \sum_j n_j$ and the total normalized energy $E = \sum_j \beta_j n_j$, being $\beta_j$ the modal propagation constants supposed equal for the degenerate modes.

The optimization process essentially aims to reach a state of thermal equilibrium for the multimode optical system, where the energy distribution among the different modes and groups is in a state of entropy extremum while satisfying the conservation constraints imposed by $N$ and $E$. This leads to

$$\frac{\partial}{\partial n_l}\left[S + \sum_{j=1}^{Q}\left(a'n_j + b'\beta_j n_j\right)\right] = 0, \tag{7}$$

which provides, with no approximations

$$ln\left(1 + \frac{g_j - 1}{n_j}\right) + a' + b'\beta_j = 0; \tag{8}$$

Eq. (8) is valid for

$$\frac{n_j}{g_j - 1} = \frac{1}{\exp[-(a' + b'\beta_j)] - 1}. \tag{9}$$

We can choose $a' = \mu/(Tn_0)$ and $b' = 1/(Tn_0)$, with $n_0$ a reference number of energy packets (for example the value at the lowest tested power) and $N = \gamma n_0$. $T$ (1/m) is an optical temperature and $\mu$ (1/m) is a chemical potential. We these choices, Eq. (9) can be written as

$$n_j = \frac{(g_j - 1)}{\exp\left(-\frac{\mu + \beta_j}{Tn_0}\right) - 1}. \tag{10}$$

An alternative development of Eq. (9) consists of replacing $a'$ and $b'$ with non-factorizable constants $a$ and $b$ defined as

$$-(a + b\beta_j) = ln\left[\frac{1}{n_0}\exp\left(-\frac{\mu + \beta_j}{T}\right) - \frac{1}{n_0} + 1\right] \simeq \frac{1}{n_0}\left[\exp\left(-\frac{\mu + \beta_j}{T}\right) - 1\right]. \tag{11}$$

The approximation in Eq. (11) is valid for $|a + b\beta_j| << 1$, which is less than $10^{-5}$ in the experiments. By replacing into Eq. (9) we obtain

$$n_j = \frac{n_0(g_j - 1)}{\exp\left(-\frac{\mu + \beta_j}{T}\right) - 1}. \tag{12}$$

The particular choice of $a$, $b$, $a'$ and $b'$ conserves the system's energy and power; in fact it results, for $a'$, $b'$

$$\sum_{j=1}^{Q} a'n_j = -\sum_{j=1}^{Q} \frac{\mu(g_j - 1)}{\mu + \beta_j}, \tag{13}$$

$$\sum_{j=1}^{Q} b'\beta_j n_j = -\sum_{j=1}^{Q} \frac{\beta_j(g_j - 1)}{\mu + \beta_j}; \tag{14}$$

and

$$\sum_{j=1}^{Q} a'n_j + b'\beta_j n_j = -\sum_{j=1}^{Q}(g_j - 1) = Q - 2M. \tag{15}$$

For $a$, $b$ it also results

$$\sum_{j=1}^{Q}(a + b\beta_j)n_j = -\sum_{j=1}^{Q}(g_j - 1) = Q - 2M, \tag{16}$$

hence, the choices of $a$, $b$ or $a'$, $b'$ are equivalent in terms of power and energy conservation.

In transitioning from Equation (10) to Equation (12) by substituting $a'$ and $b'$ with $a$ and $b$, the optimization of entropy involves the use of non-factorizable multipliers that are directly associated with the eigenvalues of the modal groups. This modification means that the extrema of entropy now have local significance because both $\mu$ and $T$ are intimately connected to the specific set of $\beta_j$ and cannot be treated as separate variables. The conservation of the total power and energy of the system is maintained, much like in the global optimization approach described in Equation (10).

The solution provided by Equation (12) is well-suited for fitting experimental modal distributions as a whole, without being influenced by the power fluctuations resulting from local condensates, corresponding to local energy minima within individual modal groups.

Now, let's consider the $i$-th mode within the $j$-th group. In the case of a GRIN fiber, where there are $2M = Q(Q + 1)$ modes and polarizations, $i$ ranges from 1 to $M$. Specifically, for $j = 1$, $i = 1$; for $j = 2$, $i$ takes values 2 and 3; for $j = 3$, $i$ ranges from 4 to 6, and so on. Additionally, the degeneracy values $g_i$ are as follows: $g_i$ equals 2 for $i = 1$, 4 for $i = 2$ and $i = 3$, 6 for $i = 4$, $i = 5$, $i = 6$, and so forth, up to $2Q$.

The mean modal power fraction over two polarizations, denoted as $|f_i|^2$, is calculated as $|f_i|^2 = 2n_j/(\gamma n_0 g_i)$, where $\gamma = N/n_0$. It's important to note that $\gamma$ needs to scale with the experimental power, meaning that as you vary the experimental power, $\gamma$ should also adjust accordingly.

Now, by introducing the differential eigenvalues $\epsilon_i = \beta_i - \beta_{j=Q}$, which are referenced to the higher mode, and $\mu' = \mu + \beta_{j=Q}$, Eq. (12) yields the weighted Bose-Einstein (BE) modal distribution:

$$|f_i|^2 = \frac{2(g_i - 1)}{g_i \gamma} \frac{1}{\exp\left(-\frac{\mu' + \epsilon_i}{T}\right) - 1}. \tag{17}$$

The constraints for $\gamma$ are twofold: it should scale with the input power, and it must satisfy the conservation law $\sum_{i=1}^{M} |f_i|^2 = 1$.

Let's consider the optical powers corresponding to $N$ and $n_0$ energy packets, denoted as $P$ and $P_0$, respectively (for an optical pulse, this is equivalent to peak power). Thus, we have $\gamma = N/n_0 = P/P_0$. The

system's internal energy can be expressed as $U = -\sum_j \beta_j n_j P/N$ (in units of W/m), and the power $P$ can be written as $P = \sum_j n_j P/N$ (in units of W).

Starting from Equations (15) and (16), we can derive the following relationships:

$$\sum_{j=1}^{Q}(a + b\beta_j)n_j = Q - 2M = \frac{\mu N}{Tn_0} + \frac{1}{Tn_0}\left(-\frac{UN}{P}\right), \tag{18}$$

which provides the state equation

$$U - \mu P = (2M - Q)P_0 T. \tag{19}$$

The local extremization problem, which yields Eq. (12), does not lead to the well-known Rayleigh-Jeans (RJ) distribution[7,33] under reasonable approximations. However, from the global extremization solution, Eq. (9), along with the definitions of $a'$ and $b'$, we can obtain the following relationship:

$$|f_i|^2 = \frac{2(g_i - 1)}{g_i \gamma n_0}\frac{1}{\exp\left(-\frac{\mu' + \epsilon_i}{Tn_0}\right) - 1}; \tag{20}$$

The presence of $N = \gamma n_0$ at the denominator of Eq. (20) makes the equation suitable for fitting experimental data only for $N < 10$. Therefore, it becomes unusable for larger values of $N$. This issue can be resolved under the assumption that $|\mu' + \epsilon_i| << |Tn_0|$, which leads directly to the Rayleigh-Jeans (RJ) distribution:

$$|f_i|^2 = -\frac{2(g_i - 1)}{g_i \gamma}\frac{T}{\mu' + \epsilon_i}. \tag{21}$$

In terms of fractional power $|c_i|^2 = P|f_i|^2$ in [W], by choosing $T' = P_0 T$ (W/m), Eq. (21) provides.

$$|c_i|^2 = -\frac{2(g_i - 1)}{g_i}\frac{T'}{\mu' + \epsilon_i}. \tag{22}$$

The RJ distribution is a suitable choice for describing experiments characterized by global condensation states, such as self-cleaning experiments. On the other hand, the more general weighted BE distribution, as given in Eq. (17), is also appropriate for fitting experimental data in cases where local condensed states are achieved, identifying an accurate trend despite local power fluctuations.

### Experimental setup

Optical pulses with a wavelength of 1400 nm and a pulse width of 250 fs are generated using an optical parametric amplifier (OPA) driven by a femtosecond Yb laser operating at a repetition rate of 100 kHz. The input beam is attenuated, linearly polarized, and passed through a quarter-wave plate ($\lambda/4$) to generate a circular state of polarization (see Fig. 5 in Note D of the supplementary material). This circularly polarized Gaussian beam is then injected into variable lengths of OM4 GRIN fiber, including spans of 1 m, 830 m, and 5 km, with an initial waist ($w_0$) of approximately 13 $\mu$m. The induced beam compression factor ($C = 2z_p/(\pi\beta_0 w_0^2)$) is calculated as 0.305, based on the self-imaging period ($z_p = 0.55$ mm), $\beta_0 = 2\pi n_0/\lambda$, and the core refractive index ($n_0 = 1.46$) of the GRIN fiber[34,35]. The effective beam waist for determining the nonlinear coefficient is $w_e = \sqrt{C}w_0 = 7.27\mu$ m[35,36], which is close to the fundamental mode waist. The use of a circular state of polarization at the input minimizes power exchanges between polarizations. The input beam is laterally shifted with respect to the fiber axis by 10 $\mu$m, in order to increase the proportion of higher-order modes.

Linear losses were measured as $\alpha = 6.0 \times 10^{-4}$ m$^{-1}$. According to Gloge theory[37], bending losses remain negligible up to the first 10 mode groups ($\alpha_j <= 8 \times 10^{-10}$ dB/km for group $j = 1, 2, \ldots, 10$, and

$\alpha >> 30$ dB/km for group $j >= 11$); in agreement with the theory, 10 modal groups could be observed at the output of both 830 m and 5 km fiber spans.

Modal dispersion is responsible for the time delay among the different mode groups; the measured delay among groups is 206 ps over 830 m, and 850 ps over 5 km. Hence, mode groups are time-resolved at the output, and easily measurable after 830 m of GRIN fiber. The time delay among groups was found to vary with distance as $\Delta t_j = 1.02z^{0.79}$; RMC affecting the experiment is intermediate between weak (where modal delay scales with $z$) and strong regime (where it changes with $\sqrt{z}$)[10].

At the fiber output, the near-field is imaged on an InGaAs camera (Hamamatsu C12741-03); the beam is also directed to a real-time multiple octave spectrum analyzer with a spectral detection range of 1100–5000 nm (Fastlite Mozza). The output pulse instantaneous power is detected by a fast photodiode (Alphalas UPD-35-IR2-D) and a real-time oscilloscope (Teledyne Lecroy WavePro 804HD) with 30 ps overall response time. An intensity autocorrelator (APE pulseCheck 50) with femtosecond resolution is also used for the temporal characterization of the input pulses. Input and output power are measured by a power meter with $\mu$W resolution.

Traditional 2D modal decomposition methods[38] are not suitable for the analysis of the output near-field after hundreds of meters of pulse propagation, because they do not account for: (i) the phase chirp which is induced by chromatic dispersion of pulses carried by the different modes; (ii) the phase delay among modes due to the modal dispersion; (iii) laser-induced phase noise; (iv) random phase differences among modes, introduced by the RMC. In this work, we use the 3D modal reconstruction method proposed in[24]; the mode group power is measured from the instantaneous power detected by a fast photodiode; samples are collected with constant delay of 206 ps, starting from the first peak; a tolerance of $\pm 25$ ps is allowed when finding the peak around the theoretical modal delay, to account for the photodiode's timing jitter. 3D fields are reconstructed with the help of coupled-mode GNLSE simulations[39], as described in ref. 24. Comparison to the measured near-field is performed after 3D reconstruction. Such method has provided an accurate estimate of the modal distributions reached in long spans of GRIN, both in linear and nonlinear regime, up to the 10-th modal group (55 modes per polarization).

### Power-flow numerical model

In the linear regime, RMC is properly modeled by the well-known power-flow diffusive equations. If $P_j$ is the power of the $j$-th mode group, the power exchange among adjacent modal groups is described by (see ref. 12)

$$P_j(z + L_c) = \left(\frac{DL_c}{\Delta m^2} - \frac{DL_c}{2m\Delta m}\right)P_{j-1}(z) + \left(1 - \frac{2DL_c}{\Delta m^2} - (\alpha_0 + Am^2)L_c\right)P_j(z)$$
$$+ \left(\frac{DL_c}{\Delta m^2} + \frac{DL_c}{2m\Delta m}\right)P_{j+1}(z), \tag{23}$$

with $m(j) = j - 1$, $D$ and $\alpha_0$ (1/m) the coupling coefficient and linear losses, respectively, $A$ the modal loss coefficient, $\Delta m = 1$ the modal step and $L_c$ the RMC integration step. In the model of Eq. (23), the first term describes the power coupled from modal group $j - 1$ to group $j$, the third term the power from $j + 1$ to $j$. Power flows in both directions, from group $j$ down to group $j - 1$ and up to $j + 1$; after consecutive integration steps of $P_j$, $j = 1, \ldots, Q$, a cascading effect is produced, causing the coupling of non-adjacent groups. However, since the weight of the first term is different from the third, the net power flow promotes the lower-order groups if higher-order modes are populated at the input end. The distance $z_{SSD}$ over which a steady-state mode power distribution is observed, can be generally calculated using the power-flow model.

Mode coupling into groups is neglected, because it is so fast that statistical modal equipartition into groups can be assumed; it is then meaningful to calculate the mean modal content into groups as $|f_i|^2 = 2P_j/(g_i P_{tot})$.

**Modal power equipartition.** As a consequence of Eq. (23), the power flows asymmetrically between different modal groups of multimode fibers. It is possible to demonstrate, using the alternative Gibb's definition of entropy[40], that modal power equipartition does not generally apply among different modal groups. Given, at steady state, $p_i = <|f_i|^2>$ the modal occupation probability, and $\lambda_1, \lambda_2$ two Lagrange multipliers, extremization of the entropy, while power $P$ and energy $E$ are conserved, reads

$$\frac{\partial}{\partial p_k}\left[-\sum_{i=1}^{M} p_i \log p_i + \lambda_1\left(\sum_{i=1}^{M} p_i - 1\right) + \lambda_2\left(\sum_{i=1}^{M} p_i \beta_i - E/P\right)\right] = 0; \quad (24)$$

from Eq. (24), if only the power is conserved ($\lambda_2 = 0$), we obtain

$$\sum_{i=1}^{M} p_i = \sum_{i=1}^{M} \exp(\lambda_1 - 1) = M \exp(\lambda_1 - 1) = 1, \quad (25)$$

which brings to the modal equipartition $p_i = 1/M$. However, when considering also the conservation of the energy, Eq. (24) provides

$$\sum_{i=1}^{M} p_i = \sum_{i=1}^{M} \exp(\lambda_1 + \lambda_2 \beta_i - 1) = 1, \quad (26)$$

which provides equipartition only for degenerate modes, with same $\beta_i$.

**GNLSE numerical model**
In the nonlinear (and linear) regime, numerical simulations use the coupled GNLSEs[39], modified to include modal and wavelength-dependent losses, and linear random-mode coupling (RMC) for mode $p$

$$\frac{\partial A_p(z,t)}{\partial z} = i\left(\beta_0^{(p)} - \beta_0\right)A_p - \left(\beta_1^{(p)} - \beta_1\right)\frac{\partial A_p}{\partial t} + i\sum_{n=2}^{4}\frac{\beta_n^{(p)}}{n!}\left(i\frac{\partial}{\partial t}\right)^n A_p - \frac{\alpha_p(\lambda)}{2}A_p$$
$$+ i\sum_m q_{mp}A_m + in_2 k_0 \sum_{l,m,n} S_{plmn}\left[(1-f_R)A_l A_m A_n^* + f_R A_l\left[h_R^*\left(A_m A_n^*\right)\right]\right]. \quad (27)$$

In Eq. (27), $\beta_n^{(p)}$ is the n-th order dispersion term (modal and chromatic) for mode $p$, $\alpha_p(\lambda)$ the modal and wavelength-dependent loss coefficient, $n_2$ (m²/W) the nonlinear index coefficient multiplying the Kerr and Raman terms, $S_{plmn}$ is an overlap integral among modes, accounting for IM-FWM, and $q_{mp}$ is the linear RMC coupling coefficient, from mode $m$ to $p$, coming from the power-flow equations model[12]

$$q_{mp} = \begin{cases} \left[\frac{DL_c}{p-1}\left(1 - \frac{1}{2(p-1)}\right)\right]^{1/2} & \text{from modes } m \text{ with } g_m = g_p - 1 \\ \left[\frac{DL_c}{p+1}\left(1 + \frac{1}{2(p-1)}\right)\right]^{1/2} & \text{from modes } m \text{ with } g_m = g_p + 1, \end{cases} \quad (28)$$

being $g_p$ the degeneracy of modes, $p = 1, 2, \ldots, M$, $D$ (m⁻¹) the RMC coupling coefficient, and $L_c$ the RMC numerical step. Degenerate modes are not accounted for in Eq. (28), because their coupling is so fast that power equipartition can be assumed into groups. The model of Eq. (28) holds for single-wavelength transmission; in wavelength-division multiplexed systems (WDM), a more complex model should be adopted.

In the main text, fiber parameters at $\lambda = 1400$ nm are: $\beta_2 = -11.8$ ps²/km, $\beta_3 = 0.102$ ps³/km for the fundamental mode, $\alpha = 2.6$ dB/km, negligible modal loss, $n_2 = 2.7 \times 10^{-20}$ m²/W, index parabolic factor $g = 2.08$, relative index difference $\Delta = 0.010$, core radius $a = 25\mu m$,

$D = 0.003$ (1/m), $L_c = 6$ mm, $f_R = 0.18$, $S_{plmn}$ calculated from modal overlap integrals.

## Data availability
The datasets generated during and/or analysed during the current study are available from the corresponding author on request.

## Code availability
Not applicable

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

## Acknowledgements

The authors wish to thank C.Conti and G.D'Aguanno for valuable discussions about glass states, S. Savovic' for information regarding the power-flow model, V.Yankov, M.Ferraro, D.Christodoulides, G.Pyrialakos, A.Picozzi and G.Steinmeyer for discussion regarding optical thermodynamics, and F.Wise and L.Wright for making freely available the open-source parallel numerical mode solver for the coupled-mode nonlinear Schrödinger equations[41], and the test data used in Note A of the supplementary material; a modified version of the software was used for the numerical simulations. Project ECS 0000024 Rome Technopole, Funded by the European Union - NextGenerationEU (M.Z.). HORIZON EUROPE European Research Council (101081871) (S.W.).

## Author contributions

M.Z. and F.M. developed the theory. M.Z. conducted the simulations and the experiments. S.W. funded and managed the project. All the authors contributed in preparing the manuscript.

## Competing interests

The authors declare no competing interests.
