## [Peer Review File · Nature Communications]

Statistics of modal condensation in nonlinear multimode fibersREVIEWER COMMENTS

Reviewer #1 (Remarks to the Author):

This manuscript reports on the propagation of incident pulses comprised of multiple transverse modes and at various pulse energies into a disordered multimode GRIN fiber. The modes are mixed by disorder and nonlinear effects. As the pulse energy and fiber length increase, the pulse evolves through a system with subsets of coupled modes towards a single soliton. This is described in terms of weighted Bose-Einstein condensation,

The abstract mentions the use of "a new 3D mode decomposition method to obtain" the modal distribution of the pulse within the fiber "with unprecedented accuracy,". It isn't apparent what is fundamentally new about the method employed or how the accuracy is superior to standard accuracy. Important context is missing, such as a discussion of the advance made in this article relevant to previous work, such as in Ref. 8.

The authors mention the relevance of this work to SDM. But this needs to be explained since MM fibers generally are nearly free of scattering. Over what length scales in standard fibers does the joint action of scattering by disorder and nonlinearity become important. Is this at all a concern in telecommunications. There is no figure of merit provided regarding the scattering strength such as the mean free scattering length of the scattering time. These issues should be addressed explicitly before one can make a firm judgement on the appropriateness of the article for Nature Communications.

Before the paper is published in any journal, the grammar should be improved. There are grammatical errors and nonstandard expressions in every few sentences. Much of this could be handled with judicious use of CHAT-GPT or a paid scientific editor. Below I list a few points through the first page of the Introduction that give a sense of the problem as I see it. I won't go further since, once pointed out, the problem can be solved.

- There should be a space between Fig. and the number of the figure.
- A sentence should not start with Fig., it should be Figure...
- “On the other hand,” should not be used for “And.” It should present a contrast between two statements.
- Because there are so many acronyms used, it might be useful to spell some of these out throughout the article.
- There are items that are not explained such as the word “chaotic” in the first sentence of the abstract, the meaning of “bullet” in the phrases “3D bullets,” “soliton bullet,” and “optical bullets.”
- In the second sentence of the abstract, by “can obtain steady states,” do the authors mean “can reach steady state”
- the authors mention “optical glass.” Do they mean a quasi-steady state in which the lowest temperature is not reached so that all the modes are not mixed to create a soliton?
- “statistical mechanics permits to estimate the equilibrium distribution of these packets among degenerate groups of modes.” could be “the equilibrium distribution of these packets among degenerate groups of modes can be estimates with use of statistical mechanics”
- “grows large \diamond “increases”
- “ At telecom wavelength” \diamond At the telecom wavelength. The word “the” is often missing.
- “Pulses corresponding to the different modal groups get shorter and increase their peak power” \diamond “Pulses corresponding to different modal groups become shorter and their peak power increases” – note, the word “the” should be removed where it does not belong.
- “pulsewidth,” should be two words “pulse width”
- “a quasi-soliton forms, characterized by pulses carrying the individual modal groups which are initially overlapped in time, and reduce their pulsewidth to values approaching the soliton TFWHM” \diamond “a quasi-soliton forms, characterized by pulses carrying the individual modal groups, which initially overlap and with pulse width that approach the soliton linewidth.”
- “while Raman self-frequency shift is not able to affect the pulse wavelength yet; a conservative

wave propagation system can still be assumed." ◊ " The pulse energy is conserved as long as the wavelength is not shifted by Raman scattering"

Reviewer #2 (Remarks to the Author):

The manuscript "Statistics of modal condensation in nonlinear multimode fibers" reports on measurements of the modal decomposition at the output of a nonlinear multimode optical fiber with random mode coupling. The authors compare the experimental results with a thermodynamic theory based on a weighted Bose-Einstein distribution and numerical investigation of the linear power-flow equations and the nonlinear complex Ginzburg-Landau equation that models the multimode fiber system. The results might confirm a steady state for long fibers (800m) and that the weighted Bose-Einstein distribution is a reliable approach to analyze such a steady state in linear and nonlinear regimes.

The results reported in the manuscript are interesting and may provide another step towards a more complete understating of the subtle interplay between nonlinearity, mode coupling, and modal interference. However, I doubt the rigidity of the conclusions drawn for the reported experimental and numerical results meets the criteria for publication in Nature Communications. As I explain in more detail below, many claims throughout the paper are not supported by convincing evidence. Some of the statements are too vague to identify the novelty of the presented results concerning previous works. Therefore, I cannot recommend publishing this work in Nature Communication.

My main concern is that I do not think the authors proved the weighted Bose-Einstein (wBE) distribution is indeed a suitable description for the steady-state solution in both the linear and nonlinear regimes:

1. I do not understand how a linear system can thermalize. A linear system is integrable; A system with N normal modes has N constants of motion that depend on the initial conditions. How can it thermalize? I appreciate that the 'modes' analyzed in this work are the 'guided modes' of a perfect fiber and not the 'normal modes' of a fiber with random mode mixing. But doesn't that mean the modal power fraction will keep changing along the fiber and not thermalize? In explaining how one can observe thermalization in a linear system, please also explain the thermalization length, i.e., the fiber length after which one observes "thermalization," and whether the thermalization depends on the exact model for the random mode mixing.

2. The results in the linear regime are explained by the claim that "the mode coupling coefficients are not symmetrical in the two directions, resulting into a preferential transfer of power from HOMs into low-order modes of the fiber." I do not understand this statement – doesn't it violate optical reciprocity? I can appreciate that, for example, the coupling from mode 2 to mode 1 is higher than from mode 2 to mode 3. But from the optical reciprocity theorem, the coupling from mode 2 to mode 1 should be identical to that from mode 1 to mode 2. Please explain what "not symmetrical in the two directions" means and how it can yield power flow to low-order modes. Does one have to assume mode-dependent loss to get this breaking of symmetry between the high and low order of modes in the linear regime?

3. The experimental data is based on the modal decomposition method described in figure 2, where the relative height of the peaks, marked by the red circles, are associated with the group power fraction. It is, therefore, critical to identify the correct peak values, but it is not clear what is the procedure to extract these values. The text mentions, "...we sample the photodiode traces at points corresponding to the relative group delay of each modal group." What does that mean? Does one assume a value for the group delay for a perfect fiber and then register the power at fixed delays from the main peak, assuming some fiber length? But then, I would expect the same differential group delays for the linear and nonlinear regime; this does not seem to be the case. It is also clear that red circles miss some peaks, for example, in figure 2c. Please explain in detail the protocol for sampling the photodiode samples.

4. In the intermediate regime, on the one hand, the authors say that "the modes assume a

characteristic distribution, which cannot be traced back to what has been observed in the linear or soliton regimes ." On the other hand, it is claimed that "Our results extend the thermodynamic approach to multimode fibers to unexplored optical states, which acquire the characteristics of optical glass." I do not understand from the two statements whether wBE captures the intermediate regime or not. If it does, and indeed in all regimes the distribution follows wBE, what does it mean the distribution cannot be traced back to what has been observed in the linear and soliton regimes?

5. I was expecting to be able to assess by myself whether wBE captures the intermediate regime. However, the reported results are not detailed enough for that. First, the exact fitting procedure is not explained well. For example, it is unclear whether γ was obtained independently as a fitting parameter for each data set reported in Table 1 or if the same parameter was used for all data sets since E_{in}/γ should be constant. I would expect the latter approach to avoid overfitting.

There is also no figure of merit to assess the goodness of the fits.

It would have also been helpful to plot the data and curves in panels 3a,b,c, and d on the same plot. For example, it is hard to compare 3a and 3b; the trend of the experimental data looks quite similar in the curves. But the text accompanying these figures describes different trends (straight line in a versus higher power in mode group 2 when describing figure 3b). But this is evident also in figure 3a. So, is the higher occupation in mode group 2 also a feature of the linear system? It does not seem like the case in the numerical simulation (figure 4b). If there is still some nonlinearity in 3a, why not decrease the power even more?

In addition, I would like to raise some more technical points:

6. It is claimed that "the weighted BE is able to fit the output modal distributions (not shown) obtained from the simulations of Fig.4b, denoting the achievement of steady states." This sounds like a powerful and important result that can help the validity of wBE. Why not show these fits to the numerical simulations?

7. In figure 6a, what is the role of random mode mixing? What would change in the figure if one turns off mode mixing? What happens in the strong mode mixing regime? How were the values for D and L_c chosen? Why is it different than the values selected in section 4.1, figure 5?

8. I do not understand the statement "the linear and nonlinear interaction among different modal groups is mostly concentrated in the first portion of the fiber, where pulses overlap in time ." What does it mean 'linear interaction'? How can linear effects be confined to the first part of the fiber? Intuitively, I would expect linear effects to be significant throughout the entire propagation of the fiber. Please clarify.

9. The authors should explain what they mean by 'glass states' – the term is mentioned as a selling point in the abstract, introduction, and conclusions – but it is never defined, and it is not explained in what sense the intermediate regime acts as a glass state.

10. The authors mention they also measure 1m and 5km long fibers, but as far as I understand, the reported results are only for 830m. What were the measurements with 1m and 5km fibers used for?

11. The figures' captions do not provide enough details to understand the figures.

12. How are β_j and ϵ_j computed? What are the assumptions on the index profile of the fiber? Is mode mixing taken into account in this computation?

Reviewer #3 (Remarks to the Author):

This manuscript presents very interesting results on experiments and models of multimode fibers. Both experiments and numerics are highly involved and the procedures followed are the result of recent studies by these authors and groups they have interacted with.

1. From the title, abstract and introduction it was not immediately obvious to this reader what possible implications the results may have for applications of multimode fibers. Stepping back from the intricacies of the statistical perspective to a simpler practical one may be helpful for many readers.

2. Progressing through the manuscript, it emerged that a critique of refs 28 and 22 and possible generalization of their results is eventually presented. The authors do not mention this in the introduction; they wait till Appendix A to present this aspect of the study. Is this an incorrect reading of the manuscript? The authors may consider extending the introduction to address this point.

The manuscript is well written and organized, despite the complexity of the material.

Reviewer #4 (Remarks to the Author):

Multi-mode optical fiber has been extensively studied in the field of communication as a transmission medium. Most of the interactions between modes are based on the mode-coupling theory. There are also some experimental results that describe the intensity distribution of the interaction after it enters a stable state, but there is no mature theoretical method for the complex process of interaction between modes. From this perspective, this paper is meaningful for related researchers. In addition, the application of multi-mode fiber as a scattering medium in the field of imaging has received special attention in recent years. The influence of the interaction between modes on the evolution of the coherent performance of the laser beams also requires the development of similar theoretical models. Therefore, it is more significant to develop a generalized theoretical model for similar problems.

The authors have studied beam propagation in multi-mode fibers in the presence of linear random mode coupling and Kerr nonlinearity, and introduced a weighted Bose-Einstein law to describe steady-state modal power distributions both in the linear and nonlinear regimes. I think the idea and results are interesting for the readers in related fields.

There are some points I suggest the authors to revise:

1 please explain the acronyms when they are first time appeared, such as IM-FWM;

2 please provide the experimental setups, better with photos, which matters to the obtained results.

3 If the wavelength is within the visible range, the simulation and experimental results will be similar ??

I suggest this paper to be accepted for publication with major revisions.

Point-to-point response to the Reviewers' Comments

Reviewer #1 (Remarks to the Author):

This manuscript reports on the propagation of incident pulses comprised of multiple transverse modes and at various pulse energies into a disordered multimode GRIN fiber. The modes are mixed by disorder and nonlinear effects. As the pulse energy and fiber length increase, the pulse evolves through a system with subsets of coupled modes towards a single soliton. This is described in terms of weighted Bose-Einstein condensation,

The abstract mentions the use of “a new 3D mode decomposition method to obtain” the modal distribution of the pulse within the fiber “with unprecedented accuracy,”. It isn't apparent what is fundamentally new about the method employed or how the accuracy is superior to standard accuracy. Important context is missing, such as a discussion of the advance made in this article relevant to previous work, such as in Ref. 8.

ANS: We thank the Reviewer for the precious comments. The 3D mode decomposition method has been described in detail by the authors in [23]. From the photodiode's traces, it is possible to extract accurate measurements of the power carried by the fundamental and higher-order modes; a 3D reconstruction (output near field and instantaneous power) is possible using simulations with coupled-mode GNLSE models, compared to experimental photodiode traces and output near-field, as explained in [23]. The method is necessary in long spans of multimode fiber to overcome the difficulties arising from (i) the chromatic dispersion-induced phase chirp of pulses carried by different modes; (ii) the mode dispersion-induced group delay among modes; and (iii) the phase randomization caused by the random-mode coupling (RMC).

Common 2D modal decomposition methods are effective when phase coherence is conserved across the pulse duration; however, for fiber lengths above a fraction of the dispersion length and/or above the RMC correlation length, these methods cannot be used, because of the phase incoherence induced by the effects listed above.

In order to stress this point, we added on Sec 6.2: “samples are collected with ...3D fields are reconstructed with the help of coupled mode GNLSE simulations [40] as described in [23]”

The novelty of this article respect to previous work is briefly summarized here:

- A new weighted Bose-Einstein distribution is theoretically demonstrated; experiments and simulations show its accuracy in describing thermalized states in comparison with the well-known Rayleigh-Jeans distribution.
- The weighted BE law is experimentally and numerically found suitable to describe also the linear regime steady-state distributions (SSD) induced by RMC.
- Experiments and simulations demonstrate the thermalization of multimode solitons in long spans of GRIN fiber.
- It is also demonstrated the existence of an intermediate regime, defined as quasi-soliton, where local condensation of energy is produced in the lower-order modal groups (2-4), different from the ground state. The observed distributions describe glass states of the optical field, similar to those already observed in disordered lasers and Bose-Einstein condensates.
- We introduce in Sec. 6.4 a new RMC model to be used with coupled GNLS equations.

- We demonstrate in Sec. 6.3.1 the impossibility of obtaining modal power equipartition, if both system's power and energy are to be conserved.
- We describe, we believe for the first time, a new fission mechanism for multimode solitons, arising from an interplay of modal dispersion, RMC and inter-mode four-wave mixing (IM-FWM).

To better point out the above aspects, we modified in Sec. 1: "The weighted BE will prove ...highly condensed soliton state."

Section 5 was corrected, listing the novelties of this article.

The authors mention the relevance of this work to SDM. But this needs to be explained since MM fibers generally are nearly free of scattering. Over what length scales in standard fibers does the joint action of scattering by disorder and nonlinearity become important. Is this at all a concern in telecommunications. There is no figure of merit provided regarding the scattering strength such as the mean free scattering length of the scattering time. These issues should be addressed explicitly before one can make a firm judgement on the appropriateness of the article for Nature Communications.

ANS: We thank the Reviewer for raising this point. The effects of disorder are relevant on a fraction of the steady-state distance z_{SSD} , where a steady-state, non-thermalized modal distribution is reached. z_{SSD} has no analytical definition, and is generally calculated by using the power-flow model described in section 6.3. For a usual value $D=0.01$ [1/m] of the coupling coefficient in OM4 GRIN fibers, the z_{SSD} is less than 1 km; hence, few-mode fibers are required for telecom applications. Nevertheless, the weighted BE distribution, power-flow and GNLSE models are applicable to FMFs too, making them of interest for SDM design.

Numerical simulations of Fig. 6 show that a 100 m span of OM4 GRIN fiber is sufficient to generate a train of solitons, as a result of the interplay of modal dispersion, RMC and IM-FWM; if RMC was absent, most of the time-resolved mode group pulses would not include the fundamental mode, and soliton condensation to the ground state would be prevented.

In order to better explain the process, we added in Sec. 4.2: "The observation of a new fission mechanism in Fig. 6c ...optical fiber systems for various applications."

In Sec. 6.3 we extended the description of the power-flow model and z_{SSD} .

Before the paper is published in any journal, the grammar should be improved. There are grammatical errors and nonstandard expressions in every few sentences. Much of this could be handled with judicious use of CHAT-GPT or a paid scientific editor. Below I list a few points through the first page of the Introduction that give a sense of the problem as I see it. I won't go further since, once pointed out, the problem can be solved.

- There should be a space between Fig. and the number of the figure.
- A sentence should not start with Fig., it should be Figure...
- "On the other hand," should not be used for "And." It should present a contrast between two statements.
- Because there are so many acronyms used, it might be useful to spell some of these out throughout the article.
- There are items that are not explained such as the word "chaotic" in the first sentence of the abstract, the meaning of "bullet" in the phrases "3D bullets," "soliton bullet," and "optical bullets."

- In the second sentence of the abstract, by “can obtain steady states,” do the authors mean “can reach steady state”
- the authors mention “optical glass.” Do they mean a quasi-steady state in which the lowest temperature is not reached so that all the modes are not mixed to create a soliton?
- “statistical mechanics permits to estimate the equilibrium distribution of these packets among degenerate groups of modes.” could be “the equilibrium distribution of these packets among degenerate groups of modes can be estimates with use of statistical mechanics”
- “grows large “increases”
- “ At telecom wavelength” At the telecom wavelength. The word “the” is often missing.
- “Pulses corresponding to the different modal groups get shorter and increase their peak power”
“Pulses corresponding to different modal groups become shorter and their peak power increases”
- note, the word “the” should be removed where it does not belong.
- “pulsewidth,” should be two words “pulse width”
- “a quasi-soliton forms, characterized by pulses carrying the individual modal groups which are initially overlapped in time, and reduce their pulsewidth to values approaching the soliton TFWHM”m “a quasi-soliton forms, characterized by pulses carrying the individual modal groups, which initially overlap and with pulse width that approach the soliton linewidth.”
- “while Raman self-frequency shift is not able to affect the pulse wavelength yet; a conservative wave propagation system can still be assumed.” “ The pulse energy is conserved as long as the wavelength is not shifted by Raman scattering”

ANS: We thank the Reviewer for the precious advice. We have performed all of the corrections listed above as well as others, after careful grammar review.

The definition of optical glass has been added in Sec. 1: “In this context, a glassy state refers to a condition where the energy becomes localized within modal groups distinct from the ground state. This condition will manifest itself as an intermediary state between the low-energy disordered state and the high-energy, highly condensed soliton state.”

Reviewer #2 (Remarks to the Author):

The manuscript "Statistics of modal condensation in nonlinear multimode fibers" reports on measurements of the modal decomposition at the output of a nonlinear multimode optical fiber with random mode coupling. The authors compare the experimental results with a thermodynamic theory based on a weighted Bose-Einstein distribution and numerical investigation of the linear power-flow equations and the nonlinear complex Ginzburg-Landau equation that models the multimode fiber system. The results might confirm a steady state for long fibers (800m) and that the weighted Bose-Einstein distribution is a reliable approach to analyze such a steady state in linear and nonlinear regimes.

The results reported in the manuscript are interesting and may provide another step towards a more complete understanding of the subtle interplay between nonlinearity, mode coupling, and modal interference. However, I doubt the rigidity of the conclusions drawn for the reported experimental and numerical results meets the criteria for publication in Nature Communications. As I explain in more detail below, many claims throughout the paper are not supported by convincing evidence. Some of the statements are too vague to identify the novelty of the presented results concerning previous works. Therefore, I cannot recommend publishing this work in Nature Communication.

My main concern is that I do not think the authors proved the weighted Bose-Einstein (wBE) distribution is indeed a suitable description for the steady-state solution in both the linear and nonlinear regimes:

ANS: We thank the Reviewer for appreciating our manuscript, and for providing useful comments for its improvement.

1. I do not understand how a linear system can thermalize. A linear system is integrable; A system with N normal modes has N constants of motion that depend on the initial conditions. How can it thermalize? I appreciate that the 'modes' analyzed in this work are the 'guided modes' of a perfect fiber and not the 'normal modes' of a fiber with random mode mixing. But doesn't that mean the modal power fraction will keep changing along the fiber and not thermalize? In explaining how one can observe thermalization in a linear system, please also explain the thermalization length, i.e., the fiber length after which one observes "thermalization," and whether the thermalization depends on the exact model for the random mode mixing.

ANS: We apologize if the previous version of our manuscript could in some way suggest that thermalization may also occur in the linear regime. Indeed, an unperturbed linear system cannot thermalize: nonlinearity, hence non-integrability is necessary in order to introduce chaotic trajectories in the dynamical phase space of the system, so that all possible states are explored, which is a pre-requisite for thermalization to occur.

In the present manuscript, as well as in our previous papers [A,B], we only discuss the occurrence of thermalization in the nonlinear regime, caused by the inter-modal four-wave mixing (IM-FWM); when thermalization is reached, the modal distribution can be described by a RJ or wBE distribution. This was shown in the experiment of Fig. 3d, where the soliton regime can be considered as a condensed (thermalized) state, and by the independent tests of Figs. A1 and A2, describing thermalized self-cleaned distributions in the normal dispersion regime; in the figures, the RJ fits the experimental data up to the 8-th modal group and the wBE up to the 17-th.

To the contrary, in the purely linear propagation regime, beams do not spontaneously evolve toward an RJ thermalization. Nevertheless, a linear system including imperfections, which cause random disorder, is non-integrable as well, and it evolves towards a steady-state output mode power

distribution (SSD). RMC alone is able to produce, similarly to FWM, an ergodic average SSD, whose shape depends on the input modal distribution. The SSD can be numerically calculated by using the power-flow model of Eqs. 23 [12], and in some cases it is provided analytically; for example, when a Gaussian beam is coupled at the input, the output SSD is given by Eq. 3 in [C]; other analytical solutions are provided in [13]. In the manuscript, we reported our experimental and numerical observations, showing the occurrence of a linear SSD; we found that the wBE, characterized by three fitting parameters instead of two, is sufficiently adaptable to fit the linear SSD up to the 8-th modal group, as it is illustrated by way of example in Fig. 5b; however, this does not necessarily entail the achievement of a “thermalized state”.

In order to better explain the points above, we added:

On Sec. 4.1: “In multimode optical fibers, after a certain propagation distance represented by z_{SSD} , RMC alone can produce a steady-state output mode power distribution (SSD)... and it will be influenced by the initial power distribution among modes.”

We replaced the Fig. 5 with a simulation of the experiment of Fig. 2; we provide a direct comparison with the experimental output modal distribution in the linear regime, with 0.16 nJ pulse energy. We modified the caption “a) Results of the power-flow simulation in the linear regime, and (b) corresponding input and output mean modal power fractions vs. the modal eigenvalues, compared to the experiment. ... distribution in the linear regime.

We added the sentence “The simulation and the experiment of Fig. \ref{fig:Fig6} provide valuable insights into the power distribution and mode coupling dynamics in the multimode optical fiber. ... This demonstrates the effectiveness of the weighted BE law in capturing the dynamics of mode coupling and power flow in multimode optical fibers, even in complex and nonlinear regimes.”

In Sec. 6.3: “The distance z_{SSD} over which a steady-state mode power distribution is observed, can be generally calculated using the power-flow model.”

In the abstract we wrote: “In our analysis, we introduce a weighted Bose-Einstein law, demonstrating its suitability in describing thermalized modal power distributions in the nonlinear regime, as well as steady-state mode power distributions in the linear regime.”

[A] Mangini, F., Gervaziev, M., Ferraro, M., Kharenko, D.S., Zitelli, M., Sun, Y., Couderc, V., Podivilov, E.V., Babin, S.A., Wabnitz, S., Optics Express, 30 (7), 2022.

[B] Mario Ferraro, Fabio Mangini, Mario Zitelli & Stefan Wabnitz, Advances in Physics: X, 8:1, DOI: 10.1080/23746149.2023.2228018, 2023.

[C] A. Djordjevich, S. Savovic', J. Opt. Soc. Am. B, Vol. 21, No. 8, August 2004

2. The results in the linear regime are explained by the claim that "the mode coupling coefficients are not symmetrical in the two directions, resulting into a preferential transfer of power from HOMs into low-order modes of the fiber." I do not understand this statement – doesn't it violate optical reciprocity? I can appreciate that, for example, the coupling from mode 2 to mode 1 is higher than from mode 2 to mode 3. But from the optical reciprocity theorem, the coupling from mode 2 to mode 1 should be identical to that from mode 1 to mode 2. Please explain what "not symmetrical in the two directions" means and how it can yield power flow to low-order modes. Does one have to assume mode-dependent loss to get this breaking of symmetry between the high and low order of modes in the linear regime?

ANS: We thank the Reviewer for pointing out this aspect of the linear transmission.

Optical reciprocity holds for unperturbed linear systems; if the dielectric tensor is not isotropic, or the propagation constant is not the same for all modes, or the effective index changes with distance because of fiber imperfections, or if the local index is changed by the nonlinearity, optical reciprocity fails [D].

The solution of the power-flow equations [9,13] leads to the numerical model of Eq. 23 [12, C]. The linear coupling coefficients in the model, which have been used by the Authors also in the coupled GNLSEs [42], are given by Eq. 28; the coefficient responsible for a power transfer from modal group $m-1$ to m is different from the one from $m+1$ to m , resulting in a net transfer of power to the lower order group or to the unpopulated adjacent groups.

The power transfer is cascaded to non-adjacent modal groups by iterating the coupling step. Hence, two non-symmetrical power fluxes result, promoting the lower-order modes if higher-order modes are populated at the input. This explains the reason why SSD power $P_j(z_{\text{SSD}})$ is always centered on the fundamental mode $m(1)=0$. Power equipartition is found into the groups of degenerate modes, where statistical equipartition can be assumed [9,12,13,C]. This is demonstrated in our manuscript in Sec. 6.3.1.

In order to stress this point, we added:

On section 6.3: “In the model of Eq.~\ref{eq:PowerFlow}, ... However, since the weight of the first term is different from the third, the net power flow promotes the lower-order groups if higher-order modes are excited at the input end.”

On section 6.3.1 we added: “In linear systems affected by index imperfections, which cause disorder, as well as in nonlinear systems, the optical reciprocity law does not hold \cite{R_J_Potton_2004}. ... that modal power equipartition does not generally apply among different modal groups.”

[D] R. J. Potton, *Rep. Prog. Phys.* **67** 717, 2004

3. The experimental data is based on the modal decomposition method described in figure 2, where the relative height of the peaks, marked by the red circles, are associated with the group power fraction. It is, therefore, critical to identify the correct peak values, but it is not clear what is the procedure to extract these values. The text mentions, "...we sample the photodiode traces at points corresponding to the relative group delay of each modal group." What does that mean? Does one assume a value for the group delay for a perfect fiber and then register the power at fixed delays from the main peak, assuming some fiber length? But then, I would expect the same differential group delays for the linear and nonlinear regime; this does not seem to be the case. It is also clear that red circles miss some peaks, for example, in figure 2c. Please explain in detail the protocol for sampling the photodiode samples.

ANS: All experimental photodiode traces are sampled at a constant time delay of 206 ps between modal groups, starting from the first peak; a tolerance of ± 25 ps is allowed for finding the peak around the theoretical modal delay, to account for the photodiode's timing jitter.

We added an explanation in section 6.2: “samples are collected with constant delay of 206 ps, ... photodiode's timing jitter.”

4. In the intermediate regime, on the one hand, the authors say that "the modes assume a characteristic distribution, which cannot be traced back to what has been observed in the linear or

soliton regimes ." On the other hand, it is claimed that "Our results extend the thermodynamic approach to multimode fibers to unexplored optical states, which acquire the characteristics of optical glass." I do not understand from the two statements whether wBE captures the intermediate regime or not. If it does, and indeed in all regimes the distribution follows wBE, what does it mean the distribution cannot be traced back to what has been observed in the linear and soliton regimes?

ANS: We thank the Reviewer for notifying the contradictory sentences.

Yes, the wBE captures the intermediate regime, by fitting properly the experimental distributions (see Figs. 3b, 3c), despite of the fact that local power fluctuations are observed into groups 2-4.

On Sec. 3 we already introduced the sentence "in both cases, the weighted BE fits the experimental data up to the soliton regime,".

In Sec. 1: the sentence "traced back" was replaced: "In the intermediate energy range, typically spanning from 20\% to 80\% of the soliton energy ... In this regime, the modes assume a characteristic distribution, showing local condensation of energy among the lower-order groups (Fig.\ref{fig:Fig1}b). ... It showcases the intricate and rich behavior that can arise in optical systems under specific conditions and energy regimes."

5. I was expecting to be able to assess by myself whether wBE captures the intermediate regime. However, the reported results are not detailed enough for that. First, the exact fitting procedure is not explained well. For example, it is unclear whether γ was obtained independently as a fitting parameter for each data set reported in Table 1 or if the same parameter was used for all data sets since E_{in}/γ should be constant. I would expect the latter approach to avoid overfitting.

ANS: we apologize for the lack of details. The γ factor must scale with E_{in} ; the usual method to analyze a set of data at different powers is to fit the experiment at intermediate energy, for example $E_{in}=2.06$ nJ, and find the set of parameters γ , T , μ' which better fit the data. For the other energy levels, γ is scaled with E_{in} and it used as a fixed parameter for the fit. Figures 3c and 3d clearly show the accuracy of the wBE fits in the quasi-soliton regime.

In order to provide a detailed description of the fitting procedure, we added on Sec. 3: "In particular, γ , T , and μ' are initially determined from the fit at an intermediate pulse energy. For other energy values, γ scales with the pulse energy E_{in} , while T and μ' are determined from the fits."

There is also no figure of merit to assess the goodness of the fits.

It would have also been helpful to plot the data and curves in panels 3a,b,c, and d on the same plot. For example, it is hard to compare 3a and 3b; the trend of the experimental data looks quite similar in the curves. But the text accompanying these figures describes different trends (straight line in a versus higher power in mode group 2 when describing figure 3b). But this is evident also in figure 3a. So, is the higher occupation in mode group 2 also a feature of the linear system? It does not seem like the case in the numerical simulation (figure 4b). If there is still some nonlinearity in 3a, why not decrease the power even more?

ANS: In order to improve the analysis, we added the fit figure of merit R^2 to Table 1.

In the linear regime, we expect the lower order groups to be promoted, although they do not obey a RJ law. In the Appendix B (all appendices will be proposed as a supplementary material), we added a figure with the overlapped data of Fig. 3 in linear scale; in fact, the logarithmic scale does not permit to fully appreciate the differences between linear, quasi soliton and soliton regimes.

We added on section 3: “The r-squared figure of merit for the fits increases with energy from 0.954 to 0.999, ... capturing both the thermalization process and the steady-state behavior in multimode optical fibers.

We added “Figure B1 in the supplementary Sec. B overlaps the measured power fractions in linear scale, for a direct comparison at different energy levels.”

In addition, I would like to raise some more technical points:

6. It is claimed that “the weighted BE is able to fit the output modal distributions (not shown) obtained from the simulations of Fig.4b, denoting the achievement of steady states.” This sounds like a powerful and important result that can help the validity of wBE. Why not show these fits to the numerical simulations?

ANS: Although a wBE fit of the power-flow simulations is provided in Fig. 5b, we agree with the Reviewer on the importance of adding a further benchmark for the wBE. We added in Appendix C a new figure showing the simulated output distribution after 100 m of fiber and with 28 modes; wBE fits are reported at different energy levels.

We added the text “In Fig. ... a train of solitons is obtained, with 95\% of the power condensed to the ground state. “

On Sect. 4.2, we added the sentence “These observations highlight the dynamic nature of mode power distribution in multimode fibers and how it evolves with input pulse energy. ... that lower-order modes are promoted by the RMC even in the absence of modal losses.”

7. In figure 6a, what is the role of random mode mixing? What would change in the figure if one turns off mode mixing? What happens in the strong mode mixing regime? How were the values for D and L_c chosen? Why is it different than the values selected in section 4.1, figure 5?

ANS: We agree with the Reviewer about the poor representation of the RMC in Fig. 6a. We replaced the figure with a better zoom, showing the modal disorder beneath the modal group pulses. In the simulations with coupled GNLSE, D was chosen by way of example, in order to introduce sufficient RMC to generate a train of solitons at high power. L_c was chosen by numerical convergence requirements.

In Fig. 5, where the power-flow model is compared to the experiment, $D=0.01$ [1/m] was chosen, according to what reported in [E,F] for glass GRIN fibers.

If RMC is switched-off, modal groups just walk-off and separate in time; moreover, most of the multimode pulses do not include the fundamental mode. This prevents IM-FWM from boosting all of the power into the ground state in the soliton regime. If a strong RMC is given, it produces a continuous redistribution of power from the higher to the lower-order modes. IM-FWM in that case still can boost the promotion of the lower order modes.

In order to stress this point, we added:

On Sec. 4.2: “Setting the RMC coupling coefficient and the RMC step appropriately is crucial for achieving accurate simulation results. By adjusting these parameters to match the experimental conditions and the physical characteristics of our specific fiber, permits us to obtain simulations that closely resemble real-world behavior.”

Fig. 6a was replaced.

On Sec. 4.1: “for a glass GRIN fiber” and added refs. [E,F]

[E] J. A. van Steenwijk, *App. Opt.* 22(23), 1983.

[F] S. Savovic', A. Djordjevich, *Optics and Laser Tech.* 101, 223-226, 2018

8. I do not understand the statement "the linear and nonlinear interaction among different modal groups is mostly concentrated in the first portion of the fiber, where pulses overlap in time." What does it mean 'linear interaction'? How can linear effects be confined to the first part of the fiber? Intuitively, I would expect linear effects to be significant throughout the entire propagation of the fiber. Please clarify.

ANS: We thank the Reviewer for the correction; we agree that RMC is present throughout all of the fiber length.

On Sec. 3 we replaced “While the linear interaction is indeed significant over the entire length of fiber propagation, as mentioned in Section \ref{sec:sec1}, it's important to emphasize that it does not undermine the validity of the thermodynamic approach. The thermodynamic model remains applicable and provides valuable insights even in the presence of substantial linear interactions.”.

9. The authors should explain what they mean by 'glass states' – the term is mentioned as a selling point in the abstract, introduction, and conclusions – but it is never defined, and it is not explained in what sense the intermediate regime acts as a glass state.

ANS: We thank the Reviewer for pointing out this important aspect. We refer to [20,21] for the definition of a glass state: a state of matter or fields characterized by localized energy minima different from the ground state; in our case, this corresponds to a localized energy condensation into modal groups different from the ground state. Such states were observed in optics in disordered lasers [18] and Bose-Einstein condensation [19], but, to our knowledge, they were never observed in multimode fibers nor in soliton regime.

To stress this point, we added on Sec. 1: “In this context, a glassy state refers to a condition where the energy becomes localized within modal groups distinct from the ground state. This condition will manifest itself as an intermediary state between the low-energy disordered state and the high-energy, highly condensed soliton state.”

10. The authors mention they also measure 1m and 5km long fibers, but as far as I understand, the reported results are only for 830m. What were the measurements with 1m and 5km fibers used for?

ANS: The mentioned tests at 5 km were useful, in particular, to measure the modal delay evolution and quantify the presence of a moderate RMC. These tests, also reported in [23], permitted to assess the achievement of a steady state after 830 m.

We provide here the figures from [23], showing how modal groups are partially overlapping in time, but still groups can be sampled with proper modal delay, both in linear (a, b, c) and in soliton (d, e,

f) regimes. As in the case at 830 m, the weighted BE was able to properly fit the modal distributions in both regimes.

11. The figures' captions do not provide enough details to understand the figures.

ANS: As suggested by the Reviewer, most of the captions were corrected.

12. How are β_i and ϵ_i computed? What are the assumptions on the index profile of the fiber? Is mode mixing taken into account in this computation?

ANS: We thank the Reviewer for asking to clarify this point. β_i are calculated from the numerical model [43], and compared to the analytical formula for GRIN fibers, Eq. 16 in [13], finding good correspondence. $\epsilon_i = \beta_i - \beta_{j=Q}$, as indicated in Sec. 2; it is the differential propagation constant respect to the value for the higher measured modal group ($Q=10$ in our work). The index profile is parabolic, with the parameters of an OM4 fiber.

We added, in section 2: "... calculated according to Eq. 16 in \cite{Olshansky:75}"

On section 6.4: "parabolic factor $g=2.08$, relative index difference $\Delta=0.010$, core radius $a=25 \mu\text{m}$,"

Reviewer #3 (Remarks to the Author):

This manuscript presents very interesting results on experiments and models of multimode fibers. Both experiments and numerics are highly involved and the procedures followed are the result of recent studies by these authors and groups they have interacted with.

ANS: We thank the Reviewer for the positive comments.

1. From the title, abstract and introduction it was not immediately obvious to this reader what possible implications the results may have for applications of multimode fibers. Stepping back from the intricacies of the statistical perspective to a simpler practical one may be helpful for many readers.

ANS: We agree with the Reviewer that implications of the optical thermodynamics are not yet well understood in the Community.

We added in Sec. 1: “This thermodynamic perspective offers a significant simplification for the analysis of complex multimode systems and fibers, providing a powerful tool for designing multimode fiber transmission systems affected by disorder and nonlinear modal interactions.”

2. Progressing through the manuscript, it emerged that a critique of refs 28 and 22 and possible generalization of their results is eventually presented. The authors do not mention this in the introduction; they wait till Appendix A to present this aspect of the study. Is this an incorrect reading of the manuscript? The authors may consider extending the introduction to address this point.

ANS: We apologize if the Appendix A may appear as a critique of Refs [28, 22] (now [22 and 30]), which was not intended to be the case. To the contrary, we use the independent tests from [22] because we consider them among the most accurate results available in the literature of a thermalized state, and use them to validate the weighted Bose-Einstein distribution. We found that the Rayleigh-Jeans is able to fit the data up to the 8-th modal group, and the wBE up to the 17-th. The RJ, in our opinion, is sufficiently accurate to fit the data in most applications, and it has the advantage of being easier to use, having only two fitting parameters. The wBE is more accurate, but also more complex to use, having three parameters.

To address this point, we added in Sec. 1: “In Section A of the supplementary material, we will perform a comparison between the weighted BE equation and the well-known Rayleigh-Jeans law (RJ) [7,8], particularly when analyzing thermalized states. The former, although more complex to fit, will demonstrate superior accuracy.”

On Sec. Appendix A we added: “whose experimental modal decomposition method appears particularly accurate”

The manuscript is well written and organized, despite the complexity of the material.

ANS: We are grateful to the Reviewer for the positive assessment.

Reviewer #4 (Remarks to the Author):

Multi-mode optical fiber has been extensively studied in the field of communication as a transmission medium. Most of the interactions between modes are based on the mode-coupling theory. There are also some experimental results that describe the intensity distribution of the interaction after it enters a stable state, but there is no mature theoretical method for the complex process of interaction between modes. From this perspective, this paper is meaningful for related researchers. In addition, the application of multi-mode fiber as a scattering medium in the field of imaging has received special attention in recent years. The influence of the interaction between modes on the evolution of the coherent performance of the laser beams also requires the development of similar theoretical models. Therefore, it is more significant to develop a generalized theoretical model for similar problems.

The authors have studied beam propagation in multi-mode fibers in the presence of linear random mode coupling and Kerr nonlinearity, and introduced a weighted Bose-Einstein law to describe steady-state modal power distributions both in the linear and nonlinear regimes. I think the idea and results are interesting for the readers in related fields.

ANS: We thank the Reviewer for the positive comments.

There are some points I suggest the authors to revise:

1 please explain the acronyms when they are first time appeared, such as IM-FWM;

ANS: We checked and corrected the acronyms.

2 please provide the experimental setups, better with photos, which matters to the obtained results.

ANS: A further figure with the experimental setup was added in supplementary Sec. D

3 If the wavelength is within the visible range, the simulation and experimental results will be similar ??

ANS: We thank the Reviewer for this question. In the visible range, characterized by normal dispersion, the formation of bright solitons is prevented; however, self-cleaned thermalized states are possible. At high peak powers, the mode power distribution would organize in a similar manner to what is shown in the appendix of Sec. A; hence, the fundamental mode is the mostly populated. At low powers, the lower-order modes would be favored by RMC over long spans of fiber. For short fibers and at low powers, both RMC and IM-FWM will remain negligible (provided no narrow bending is present), and the output modal distribution would resemble the input one.

I suggest this paper to be accepted for publication with major revisions.

ANS: We thank the Reviewer for the positive assessment.

Thank you very much and we look forward to hearing from you.
Sincerely.

Dr. Mario Zitelli,

On behalf of all Authors.

REVIEWER COMMENTS

Reviewer #1 (Remarks to the Author):

Statistics of modal condensation in nonlinear multimode fibers

This article presents measurement and simulations of the evolution of modes due to random bends and nonlinearity in multimode optical fibers and demonstrates the “thermalization” of the modal power distribution into a weighted Bose-Einstein distribution. This paper gives a nuanced discussion of pulse propagation at different power levels and is more detailed and compelling than Ref. [23]. The paper is of high quality. However, it seems to me that it is a close call as to whether the additional questions addressed here, and the greater clarity achieved regarding the evolution of pulses in optical fibers are sufficiently novel and important to justify publication in Nature Communications.

The authors write, “To enhance the precision of our measurements, we employ a novel 3D mode decomposition technique, enabling us to accurately characterize modal distributions over extended lengths of graded-index fiber.” But the 3D mode decomposition method has been described in detail by the authors in [23]. Since the method was described elsewhere, it would have been helpful to refer to “this recently developed method” to avoid confusion.

The extra clarifications in the text were very helpful and the grammatical errors in the original text have been corrected. There are still some errors, mostly in the added text in red.

“We demonstrate in Sec. 6.3.1 the impossibility of obtaining modal power equipartition, if both system’s power and energy are to be conserved.” This is not clear to me since energy conservation and equipartition are not opposites.

- “This condition will manifest itself as an intermediary state” - the word is “intermediate” the word “intermediary” refers to a person who acts as a link between people in order to try to bring about an agreement.

Figure 1 doesn’t clear things up. The different symbols need to be explained

“these packets may not necessarily consist of single photons; instead, they can be composed of groups of indistinguishable photons.” But we are talking of waveguide modes, the notion of photons doesn’t enter.

On page 6, “In soliton propagation.” Is not a sentence, it must be a comma at the end of the phrase, which would continue with “the Raman”

“Assuming a 5% tolerance on energy change, an 80 nm red-shift at a 1550 nm wavelength can be tolerated in experiments.” – not clear what this means. The energy change is whatever it is. Do the authors mean usefulness of results for certain applications?

“analytical fits” really this is a fit to an analytical function

Power levels should be given in each frame of Fig. 3.

“By adjusting these parameters” Adjusting these parameters

Page 16 – “unexpected behaviors” – no “s”

“Suggesting the achievement of steady states” - suggesting that steady state is reached

“Our detailed explanations provide a clear and insightful perspective on the complex interplay of physical mechanisms” best to let others judge how insightful the perspective is

Reviewer #2 (Remarks to the Author):

I thank the authors for their detailed response. Yet, I still have some reservations concerning some of the statements in the response letter and the revised manuscript:

1a. I appreciate the authors clarified in the response letter that in the linear regime the system converges to steady-state but does not thermalize. Yet I think the following sentence on the validity of the thermodynamic approach is not entirely on par with such clarification: “At low pulse energy (Figure 3a), the weighted Bose-Einstein distribution approximates a straight line, which is in good agreement with predictions obtained by numerically solving the power-flow equations [9, 12]. This agreement highlights the validity of the thermodynamic approach in describing the mode power distribution in the linear regime”. Do the authors mean that SSD highlights the validity of the thermodynamic approach, even though thermalization is not achieved? I recommend removing this sentence.

b. It is said that the initial power distribution among the modes influences the steady-state output mode power distribution (SSD). It is crucial to explicitly explain what are the conditions of the initial power distribution that can yield an SSD that follows wBE. Specifically, it is important to know if there is an initial mode distribution that will not yield a wBE distribution. Such a discussion is essential for understanding whether in the nonlinear regime, the obtained wBE results from the initial conditions or thermalization. If the latter, in the nonlinear regime wBE should be obtained for any initial condition. Is this indeed the case?

c. The authors say, “Nevertheless, a linear system including imperfections, which cause random disorder, is non-integrable as well.” While I am not an expert in dynamical systems, to the best of my knowledge, every linear system is integrable, even with imperfections. While the structure of the normal modes of the system might not be known because of the imperfections, it is still a linear system and thus a full set of normal modes and constants of motion must exist. Please explain this point.

d. The authors mention an ‘ergodic average’. What is the source of this average in the experiment and in the simulations? Are the experiments run multiple times, with different initial conditions in each run? Or is it the pulse bandwidth that is much broader than the spectral correlation bandwidth of the multimode fiber that yields disorder averaging? Or something else? Please explain.

2. First, let me comment that I do not know of any evidence of Lorentz reciprocity breaking in optical fibers, unless via a magneto-optical effect when extremely high magnetic fields are applied, or in the presence of nonlinearity [see, for example, Jalas et al. What is — and what is not — an optical isolator, Nat Phot. (2013)]. Specifically, even in the presence of mode-mixing, one cannot claim an optical fiber can be considered a non-centrosymmetric media, one of the mechanisms discussed in ref [D] for breaking reciprocity. [D] specifically mentions that reciprocity breaking requires inelastic scattering. But mode mixing in optical fibers is elastic.

Second, I stress that reciprocity does not mean that coupling from mode n to mode $n+1$ will be lower than coupling from n to $n-1$. I do not claim that in reciprocal systems the coupling cannot be asymmetric. My point is different: asymmetric coupling does not necessarily mean the power will flow to the lower-order modes. Because once the light gets to the fundamental mode ($n=1$), it will couple back to mode $n=2$, then to $n=3$, and so on. And since from reciprocity, the coupling from mode 1 to mode 2 equals the coupling from mode 2 to mode 1, light will start flowing out of mode 1. Just like in a chain of coupled pendulums with asymmetric couplings. Without mode-dependent

loss, I do not see any reason why the power will not circulate between all modes. I speculate that the asymmetric SSD distribution results from the parabolic mode-dependent-loss term ($A \cdot m^2$ in eq. 23), and not because of the asymmetric couplings. To convince me otherwise, please compare two numerical integrations of eq. 23, one with $A=0$ and the other with the same value A used in the manuscript, but with equal coupling coefficients from P_j to P_{j-1} and from P_j to P_{j+1} . I believe the former will not show SSD, and the latter will show asymmetric SSD. If I am right, in the linear regime it is the mode-dependent-loss that causes the power to concentrate in the lower order modes, not the coupling coefficients. As I mentioned above, I believe it is critical to understand the linear regime in order to make the right interpretation of the nonlinear results.

3. In Figure 2c, the horizontal gap between the 1st and 2nd red circles is much more significant than the gap between the second and third points. Can the 25ps jitter really explain this?

5. The authors did not address the question in my previous report about the fact that in Figures 3a and 3b, the measured power in mode 2 is higher than the prediction, in both figures. Moreover, the deviation from the theory seems similar in both cases. But in the text describing Figure 3a, this deviation is ignored ("At low pulse energy (Figure 3a), the weighted Bose-Einstein distribution approximates a straight line, which is in good agreement with predictions obtained by numerically solving the power-flow equations"). Whereas in the text describing Figure 3b, it is claimed that "For a pulse energy of 0.81 nJ (Figure 3b... it can be observed that group 2 has a larger energy fraction compared to the fitting equation. It will be shown later that at this power level, there is a local condensation of energy into lower groups." The reader is led to think the deviation indicates a nonlinear effect, but it was also observed in the linear regime.

Reviewer #4 (Remarks to the Author):

The authors addressed all the issues, which were raised by my last review. I think the manuscript can be accepted for publication.

Point-to-point response to the Reviewers' Comments

Reviewer #1 (Remarks to the Author):

Statistics of modal condensation in nonlinear multimode fibers

This article presents measurement and simulations of the evolution of modes due to random bends and nonlinearity in multimode optical fibers and demonstrates the “thermalization” of the modal power distribution into a weighted Bose-Einstein distribution. This paper gives a nuanced discussion of pulse propagation at different power levels and is more detailed and compelling than Ref. [23]. The paper is of high quality. However, it seems to me that it is a close call as to whether the additional questions addressed here, and the greater clarity achieved regarding the evolution of pulses in optical fibers are sufficiently novel and important to justify publication in Nature Communications.

ANS: We thank the Reviewer for the positive comments.

The authors write, “To enhance the precision of our measurements, we employ a novel 3D mode decomposition technique, enabling us to accurately characterize modal distributions over extended lengths of graded-index fiber.” But the 3D mode decomposition method has been described in detail by the authors in [23]. Since the method was described elsewhere, it would have been helpful to refer to “this recently developed method” to avoid confusion.

ANS: The sentence was replaced according to the Reviewer’s advice. We wrote:

“To enhance the precision of our measurements, we employ a recently developed 3D mode decomposition technique,…”

The extra clarifications in the text were very helpful and the grammatical errors in the original text have been corrected. There are still some errors, mostly in the added text in red.

“We demonstrate in Sec. 6.3.1 the impossibility of obtaining modal power equipartition, if both system’s power and energy are to be conserved.” This is not clear to me since energy conservation and equipartition are not opposites.

ANS: This sentence was removed from the text.

- “This condition will manifest itself as an intermediary state” - the word is “intermediate” the word “intermediary” refers to a person who acts as a link between people in order to try to bring about an agreement.

ANS: The sentence was replaced as suggested.

Figure 1 doesn’t clear things up. The different symbols need to be explained

ANS: we thank the Reviewer for the advice. Caption in Fig. 1 was extended with “Blue (yellow) bullets correspond to the propagated lower-order (higher-order) modes. The three thermodynamic states (gas, glassy, solid) are creatively illustrated in orange colour.”

“these packets may not necessarily consist of single photons; instead, they can be composed of groups of indistinguishable photons.” But we are talking of waveguide modes, the notion of photons doesn’t enter.

ANS: We replaced the sentence with “In practical applications the number of indistinguishable energy packets involve values of n_j ranging from approximately 10^5 to 10^9 , corresponding to the number of photons in a fiber modal group.”

On page 6, “In soliton propagation.” Is not a sentence, it must be a comma at the end of the phrase, which would continue with “the Raman”

ANS: the comma was added.

“Assuming a 5% tolerance on energy change, an 80 nm red-shift at a 1550 nm wavelength can be tolerated in experiments.” – not clear what this means. The energy change is whatever it is. Do the authors mean usefulness of results for certain applications?

ANS: we agree that the sentence may be misleading. We added: “the propagation constant of the soliton changes accordingly, and so does the Hamiltonian”.

“analytical fits” really this is a fit to an analytical function

ANS: we replaced “fits obtained using”

Power levels should be given in each frame of Fig. 3.

ANS: Figure 3 was replaced, indicating the energy values in each frame.

“By adjusting these parameters” Adjusting these parameters
Page 16 – “unexpected behaviors” – no “s”

“Suggesting the achievement of steady states” - suggesting that steady state is reached

ANS: Errors have been corrected.

“Our detailed explanations provide a clear and insightful perspective on the complex interplay of physical mechanisms” best to let others judge how insightful the perspective is

ANS: we thank the Reviewer. “and insightful” was deleted.

Reviewer #2 (Remarks to the Author):

I thank the authors for their detailed response. Yet, I still have some reservations concerning some of the statements in the response letter and the revised manuscript:

1a. I appreciate the authors clarified in the response letter that in the linear regime the system converges to steady-state but does not thermalize. Yet I think the following sentence on the validity of the thermodynamic approach is not entirely on par with such clarification: “At low pulse energy (Figure 3a), the weighted Bose-Einstein distribution approximates a straight line, which is in good agreement with predictions obtained by numerically solving the power-flow equations [9, 12]. This agreement highlights the validity of the thermodynamic approach in describing the mode power distribution in the linear regime”. Do the authors mean that SSD highlights the validity of the thermodynamic approach, even though thermalization is not achieved? I recommend removing this sentence.

ANS: We thank the Reviewer for the positive comments and suggestions.

We agree that the last sentence is misleading. The validity in describing the linear regime is for the Bose-Einstein distribution, as it comes directly from the fits of the experimental data and the numerical simulations. We replaced the sentence: “This agreement highlights the validity of the Bose-Einstein distribution in describing the mode power distribution in the linear regime.”

b. It is said that the initial power distribution among the modes influences the steady-state output mode power distribution (SSD). It is crucial to explicitly explain what are the conditions of the initial power distribution that can yield an SSD that follows wBE. Specifically, it is important to know if there is an initial mode distribution that will not yield a wBE distribution. Such a discussion is essential for understanding whether in the nonlinear regime, the obtained wBE results from the initial conditions or thermalization. If the latter, in the nonlinear regime wBE should be obtained for any initial condition. Is this indeed the case?

ANS: the wBE fails describing the linear regime when a short propagation distance is involved. As an example, Fig. A shows the output modal distribution when a 200 fs pulse at 1040 nm has propagated over 1.5 m of GRIN fiber, at low power.

All the experiments and numerical simulations performed so far on long distances and low power, including RMC, provide distributions which can be properly fitted by a wBE, provided the error on the state equation, Eq. 4, is within a few percent. This is an intriguing property, which has not been demonstrated in this manuscript, and deserves a subsequent dedicated work. It might come from the property that the wBE turns out to be very accurate in the nonlinear regime; hence, the error propagates more slowly than in the case of the RJ, also when reducing the power.

In the nonlinear regime and with different input conditions, all measured distributions are properly fitted by a wBE, and in this case also by a RJ, as it is expected from the theory of Sec. 6.1; what changes are the thermodynamic parameters achieved at thermalization.

To better understand this point, we added on Sec 4.1: “provided the error on the state equation ϵ_{SE} is below a few percent”.

Fig. A

c. The authors say, “Nevertheless, a linear system including imperfections, which cause random disorder, is non-integrable as well.” While I am not an expert in dynamical systems, to the best of my knowledge, every linear system is integrable, even with imperfections. While the structure of the normal modes of the system might not be known because of the imperfections, it is still a linear system and thus a full set of normal modes and constants of motion must exist. Please explain this point.

ANS: we thank the Reviewer for highlighting the unclear sentence. Indeed, linear disordered systems have been integrated in the literature, providing explicit functional form of the output distribution, but only with simplified initial conditions (see, for example, ref. [13]).

However, the sentence appears in the first answer to the Reviewer and not in the paper. In the manuscript, there is only the sentence “In linear systems affected by index imperfections, which cause disorder, as well as in nonlinear systems, the optical reciprocity law does not hold.”, which will be discussed in comment 2.

d. The authors mention an ‘ergodic average’. What is the source of this average in the experiment and in the simulations? Are the experiments run multiple times, with different initial conditions in each run? Or is it the pulse bandwidth that is much broader than the spectral correlation bandwidth of the multimode fiber that yields disorder averaging? Or something else? Please explain.

ANS: in the previous answer to the Reviewers, ergodic average was referring to an averaging with distance. This means that averaging over several experiments with different initial conditions should be equivalent, because of the ergodicity principle, to averaging over the length of a random fiber. It is a generally accepted assumption when simulating random mode coupling effects (for example, polarization mode dispersion) in long optical fibers. In our case, we observed that several experiments, performed with different input conditions at low power, provided output distributions described by a wBE (see for example ref. [23]). The experiments of this manuscript were repeated under different input coupling conditions; each test provided an output distribution repeatable and stable in time.

2. First, let me comment that I do not know of any evidence of Lorentz reciprocity breaking in

optical fibers, unless via a magneto-optical effect when extremely high magnetic fields are applied, or in the presence of nonlinearity [see, for example, Jalas et al. What is — and what is not — an optical isolator, Nat Phot. (2013)]. Specifically, even in the presence of mode-mixing, one cannot claim an optical fiber can be considered a non-centrosymmetric media, one of the mechanisms discussed in ref [D] for breaking reciprocity. [D] specifically mentions that reciprocity breaking requires inelastic scattering. But mode mixing in optical fibers is elastic.

Second, I stress that reciprocity does not mean that coupling from mode n to mode $n+1$ will be lower than coupling from n to $n-1$. I do not claim that in reciprocal systems the coupling cannot be asymmetric. My point is different: asymmetric coupling does not necessarily mean the power will flow to the lower-order modes. Because once the light gets to the fundamental mode ($n=1$), it will couple back to mode $n=2$, then to $n=3$, and so on. And since from reciprocity, the coupling from mode 1 to mode 2 equals the coupling from mode 2 to mode 1, light will start flowing out of mode 1. Just like in a chain of coupled pendulums with asymmetric couplings. Without mode-dependent loss, I do not see any reason why the power will not circulate between all modes. I speculate that the asymmetric SSD distribution results from the parabolic mode-dependent-loss term ($A \cdot m^2$ in eq. 23), and not because of the asymmetric couplings. To convince me otherwise, please compare two numerical integrations of eq. 23, one with $A=0$ and the other with the same value A used in the manuscript, but with equal coupling coefficients from P_j to P_{j-1} and from P_j to P_{j+1} . I believe the former will not show SSD, and the latter will show asymmetric SSD. If I am right, in the linear regime it is the mode-dependent-loss that causes the power to concentrate in the lower order modes, not the coupling coefficients. As I mentioned above, I believe it is critical to understand the linear regime in order to make the right interpretation of the nonlinear results.

ANS: we thank the Reviewer for raising this important point.

It is well-known that for nonlinear media, no reciprocity theorem generally holds. Reciprocity also does not generally apply for time-varying media. Let's consider a travelling pulse in a disordered linear fiber; the refractive index experienced by the pulse changes with time as it propagates. Assume that mode 1 transfers power to mode 2 at distance z_a characterized by an index n_a ; mode 2 then transfers power to mode 1 at distance z_b characterized by an index n_b . This results in a nonreciprocal exchange of power.

The exact expressions for the coupling coefficients from group $(j+1)$ to j and from $(j-1)$ to j are provided by the power-flow theory [9, 12, 13], which provides Eq. 23; the resulting coefficients promote the lower-order groups, also in the absence of modal losses.

Fig. B

As requested by the Reviewer, Fig. B reports the experimental results and power flow simulation of Fig. 5; we also added here in panel b) the simulation with no modal loss ($A=0$, Sim. no loss). Fig. Bb shows that RMC alone is able to promote the lower-order groups of modes. As the Reviewer correctly said, the presence of modal losses (Output Sim.) amplifies the effect, further reducing the power of the higher-order modes.

Figure C reports the same simulations, but with symmetric coupling coefficients. In the absence of modal losses (Sim. no loss), the output distribution tends to be more uniform with respect to Fig. B, for trivial reasons. Again, the presence of modal losses (Output Sim.) depletes further the higher-order modes, as suggested by the Reviewer.

Fig. C

In order to better show the effect of RMC alone, we replaced Fig. 5b with Fig. B, showing also the simulation for $A=0$.

We added the sentence: “Figure 5b also reports the simulation with no modal losses ($A = 0$), demonstrating that RMC alone is able to promote the lower-order modes. Modal losses eventually enhance the effect, further depleting the higher-order modes.”

In order to better explain the reciprocity failure, we added on Sec. 6.3.1:” This can be explained by considering a pulse traveling in a disordered fiber; the pulse experiences a time-varying index as it propagates, causing a reciprocity failure.”

3. In Figure 2c, the horizontal gap between the 1st and 2nd red circles is much more significant than the gap between the second and third points. Can the 25ps jitter really explain this?

ANS: In the figure, the first 4 peaks are spaced approximately 150 fs, 250 fs, and 150 fs, within the ± 25 ps tolerance.

5. The authors did not address the question in my previous report about the fact that in Figures 3a and 3b, the measured power in mode 2 is higher than the prediction, in both figures. Moreover, the

deviation from the theory seems similar in both cases. But in the text describing Figure 3a, this deviation is ignored (“At low pulse energy (Figure 3a), the weighted Bose-Einstein distribution approximates a straight line, which is in good agreement with predictions obtained by numerically solving the power-flow equations”). Whereas in the text describing Figure 3b, it is claimed that “For a pulse energy of 0.81 nJ (Figure 3b... it can be observed that group 2 has a larger energy fraction compared to the fitting equation. It will be shown later that at this power level, there is a local condensation of energy into lower groups.” The reader is led to think the deviation indicates a nonlinear effect, but it was also observed in the linear regime.

ANS: We thank the Reviewer for addressing to this point.

In Sec. 2 we explain: “Power fluctuations related to local condensation do not invalidate the equation, as it is used to fit the power distribution of the modal groups as a whole. In the following section, Equation 2 will be applied for this purpose.”

Figure 4a is more suitable to describe local power condensation into groups 2-4 at intermediate energies, which correspond to the power deviations observed in Fig. 3. It is noted an increase of energy on group 2 when passing from 0.16 nJ to 0.82 nJ, followed by a decrease when passing to 1.65 and higher energies. This effect is outside of the measurement error, and is even more clear in the simulations of Fig. 4b.

In order to provide more evidence to this property, we added on Sec. 3: “When passing from 0.16 nJ to 0.82 nJ input energy, the power of group 2 increases, followed by an important decrease for 1.65 nJ input energy or higher values.”

Reviewer #4 (Remarks to the Author):

The authors addressed all the issues, which were raised by my last review. I think the manuscript can be accepted for publication.

ANS: We thank the Reviewer for the positive assessment.

Thank you very much and we look forward to hearing from you.

Sincerely,

Dr. Mario Zitelli,

On behalf of all Authors.

REVIEWERS' COMMENTS

Reviewer #1 (Remarks to the Author):

The experimental demonstrations are in good agreement with sound analysis of modal redistribution in MM fiber. The revised paper convincingly explains and demonstrates the ability of weighted the Bose-Einstein distribution to describe the distribution of modes in random MM fiber with linear mode coupling and for fibers with four-wave mixing due to Kerr nonlinearity. Thus this work gives confidence that theoretical models of modal thermalization can be used reliably to analyze pulse propagation in optical fibers. in confirms theoretical work I therefore recommend the paper for publication in Nature Communications.

Reviewer #2 (Remarks to the Author):

I think there are several wrong statements in the paper, for example, that disorder breaks optical reciprocity. And that in a linear system, a steady state can be derived from a maximal entropy consideration (see section 6.3.1). I think it is better for the readers to be able to judge by themselves, based on the referee reports available to them. Hence I recommend publishing the paper as is. Finally I would like to share my view that might be, strongly useful in my opinion, and critical to eliminate the errors in the paper.

For the sake of completeness, I share with the authors my thoughts on these two topics. As for reciprocity, I believe the authors are mixing between time-reversal symmetry and axial symmetry. Coupling between modes may change along the propagation axis of the fiber, but the coupling does not change in time. So for monochromatic light, one might talk about the mapping of time to the propagation axis, but that must be stated and then one needs to explain why it leads to asymmetric power flow to low-order modes. As for a steady state in the linear regime, let's assume that the system is unitary, i.e. neglecting loss. Then if there is an input that leads to a flow towards a low-order mode, then there must be an orthogonal input that will lead to an orthogonal output, i.e. an output that will occupy more higher-order modes than lower-order modes. Hence one expects that on average over all inputs, all modes will be occupied.